# Comprehensive analysis of iron utilization by *Mycobacterium tuberculosis*

**Lei Zhang**[1], **R. Curtis Hendrickson**[1], **Virginia Meikle**[1], **Elliot J. Lefkowitz**[1], **Thomas R. Ioerger**[2], **Michael Niederweis**[1]*

**1** Department of Microbiology, University of Alabama at Birmingham, Birmingham, Alabama, United States of America, **2** Department of Computer Science and Engineering, Texas A&M University, College Station, Texas, United States of America

* mnieder@uab.edu

**Data Availability Statement:** All relevant data are within the paper and its Supporting Information files except for the transposon-junction sequencing data which are available from NCBI Sequence Read

## Abstract

Iron is essential for nearly all bacterial pathogens, including *Mycobacterium tuberculosis* (Mtb), but is severely limited in the human host. To meet its iron needs, Mtb secretes sidero-phores, small molecules with high affinity for iron, and takes up iron-loaded mycobactins (MBT) and carboxymycobactins (cMBT), from the environment. Mtb is also capable of utilizing heme and hemoglobin which contain more than 70% of the iron in the human body. However, many components of these iron acquisition pathways are still unknown. In this study, a high-density transposon mutagenesis coupled with deep sequencing (TnSeq) showed that Mtb exhibits nearly opposite requirements for 165 genes in the presence of heme and hemoglobin versus MBT and cMBT as iron sources. The ESX-3 secretion system was assessed as essential for siderophore-mediated iron uptake and, surprisingly, also for heme utilization by Mtb. Predictions derived from the TnSeq analysis were validated by growth experiments with isogenic Mtb mutants. These results showed that (i) the efflux pump MmpL5 plays a dominant role in siderophore secretion, (ii) the Rv2047c protein is essential for growth of Mtb in the presence of mycobactin, and (iii) the transcriptional repressor Zur is required for heme utilization by Mtb. The novel genetic determinants of iron utilization revealed in this study will stimulate further experiments in this important area of Mtb physiology.

## Author summary

Tuberculosis is caused by *Mycobacterium tuberculosis* and is the leading cause of death from a single infectious disease, resulting in approximately 1.5 million deaths per year worldwide. *M. tuberculosis* resides in granulomas which are formed in the lung in an attempt to wall off the infection. Recently, it was shown that iron-sequestering proteins accumulate in these granulomas to high concentrations, establishing an environment devoid of free iron. In this study, we examined the contribution of each gene towards the utilization of different iron sources by *M. tuberculosis*. We identified 165 genes involved in iron utilization by Mtb, 66 of which were classified as essential or as required for optimal growth of *M. tuberculosis* in the presence of different iron sources. Our study provides

Archive (www.ncbi.nlm.nih.gov/sra) under
BioProject PRJNA575878.

**Funding:** This work was supported by the grant
R01 AI137338 from the National Institutes of
Health (NIH) and an AMC21 grant from the
University of Alabama at Birmingham to MN. The
Center for Clinical and Translational Science (EJL,
RCH) is supported by the award UL1TR003096
from the National Center for Advancing
Translational Sciences of the National Institutes of
Health. DNA sequencing of the TnSeq libraries was
done by the UAB Genomics Core which is
supported by grant CA013148 from the NCI. The
funders had no role in study design, data collection
and analysis, decision to publish, or preparation of
the manuscript

**Competing interests:** The authors have declared
that no competing interests exist.

an unprecedented insight into the genetic determinants of iron utilization by *M. tuberculosis.*

## Introduction

Iron is an essential nutrient for almost all organisms because of its vital role as a redox cofactor of proteins required for cellular processes ranging from respiration to DNA replication [1,2]. In healthy humans, almost all iron is bound to proteins such as ferritin for storage, to transferrin and lactoferrin for transport or bound as a cofactor of heme in hemoglobin or in iron sulfur clusters [3]. Low iron availability limits the replication of bacterial pathogens which is further reduced by an innate immune response which sequesters iron in the human body [4,5]. To overcome iron limitation, pathogenic bacteria have evolved a variety of strategies to compete for iron in the host and to establish an infection. *Mycobacterium tuberculosis* (Mtb), the causative agent of tuberculosis (TB), utilizes two pathways to acquire iron during infection. Mtb produces small molecules with extremely high affinity for iron (III) called mycobactins (MBT) and carboxymycobactins (cMBT) [6,7]. These siderophores share a common core structure determining their iron (III) chelating properties, but the secreted cMBTs have significantly shorter fatty acid chains and are, hence, more hydrophilic than the MBTs which are completely insoluble in water and are, therefore, membrane associated [6,8]. These siderophores are capable of removing iron from host proteins such as transferrin and lactoferrin [9]. Mtb takes up the iron-bound siderophores using the IrtA/IrtB transporter [10], reductively removes iron (III) from the siderophores [11] and secretes the iron-free siderophores through the MmpL4/MmpS4 and MmpL5/MmpS5 efflux pumps [12] for a new cycle of iron uptake [13]. These siderophores could be shared among Mtb cells in areas of high bacterial loads such as in macrophages or in the lungs of patients with active TB via intracellular [14] or extracellular vesicles [15], respectively, enabling efficient iron uptake. The ESX-3 system, one of five type VII secretion systems of Mtb [16], also plays an important role in iron acquisition by Mtb [17,18,19]. However, it is not understood how the ESX-3 system contributes to siderophore-mediated iron acquisition by Mtb. While Mtb siderophores have been identified as important mycobacterial growth factors in the 1960s [20,21,22], we and others discovered recently that Mtb is also capable of directly utilizing heme and hemoglobin as iron sources [23,24]. This pathway is important because 70% of the available iron in the human host is stored in heme [25], mostly in hemoglobin, which is inaccessible to siderophores. Several proteins have been implicated in heme utilization by Mtb, such as the heme-binding protein Rv0203 [24], the RND efflux pumps MmpL11 and MmpL3 [24] and the cell surface proteins PPE36, PPE62 [26] and PPE37 [27]. However, the molecular mechanisms of how these proteins contribute to heme uptake by Mtb remain unclear. Furthermore, significant gaps remain in our understanding of both siderophore-mediated and heme-dependent iron uptake. For example, it is unclear how siderophores are secreted and taken up across the outer membrane [28] and whether Mtb possesses a TonB-like system which is essential for siderophore and heme uptake in gram-negative bacteria [29,30].

The importance of iron acquisition for virulence of Mtb has been shown in both mice infection experiments [10,31,32] and in granulomas of human TB patients [33]. In an attempt to identify novel components of the iron acquisition pathways of Mtb, we performed high-density transposon mutagenesis coupled with deep sequencing (TnSeq) to systematically assess genetic requirements for Mtb growth utilizing different iron sources. This analysis revealed that Mtb exhibits nearly opposite genetic requirements for 165 genes in the presence of heme

and hemoglobin versus MBT and cMBT as sole iron sources. One surprising exception was that the ESX-3 secretion system is essential not only for siderophore-mediated iron uptake as previously shown [17], but also for heme utilization by Mtb. Furthermore, we demonstrated that the RND efflux pump MmpL5 and Rv2047c, a protein of unknown function, are important for Mtb growth in the presence of mycobactin. We also showed that two transcriptional repressors, Zur and SmtB, play pivotal roles in heme and hemoglobin utilization by Mtb. Importantly, this work provides a comprehensive characterization of the genetic determinants of iron acquisition by Mtb.

## Results

### Iron sources utilized by *M. tuberculosis*

In order to establish a solid basis for identifying genetic requirements for iron utilization by Mtb, we first wanted to establish which iron sources Mtb can utilize. In particular, we aimed to examine the proposed utilization of transferrin and lactoferrin as iron sources [34,35] which is in contrast to our previous result [23]. To this end, we utilized the *mbtD* deletion mutant ML1600 which does not produce mycobactins and carboxymycobactins [12,23]. We measured growth of the avirulent parent strain Mtb mc$^2$6230 (H37Rv ΔRD1, Δ*panCD*; S1 Table) and of Mtb ML1600 in self-made low-iron Middlebrook 7H9 medium using 20 μM ammonium ferric citrate (control), 10 μM human holo-transferrin, 10 μM human holo-lactoferrin, 5 μM human hemoglobin, 20 μM hemin, 0.2 μM mycobactin and 0.2 μM carboxymycobactin as sole iron sources, respectively. These experiments clearly showed that Mtb utilizes iron from mycobactin, carboxymycobactin, hemin and hemoglobin consistent with previous results [23,24,27,36]. In contrast, iron salts, transferrin and lactoferrin were only used in the presence of siderophores (Fig 1). This result is consistent with previous studies establishing that Mtb siderophores can solubilize iron from transferrin or lactoferrin [37,38] indicating that Mtb does not produce a specific transferrin or lactoferrin uptake system under those growth conditions.

### Construction and characterization of Mtb transposon libraries grown with different iron sources

To reveal the genetic requirements for iron utilization by Mtb we constructed transposon libraries of Mtb Δ*mbtD* (ML1600) using a modified *himar1*-based transposon [39]. The Mtb transposon library was grown on self-made low-iron 7H9 agar plates containing 0.5% bovine albumin and 0.01% tyloxapol supplemented with the following iron sources: (i) 500 ng/mL (~0.64 μM) Fe-cMBT; (ii) 250 ng/mL (~0.275 μM) Fe-MBT ("high" MBT); (iii) 50 ng/mL (~0.055 μM) Fe-MBT ("low" MBT); (iv) 20 μM hemin; (v) 5 μM hemoglobin and (vi) 20 μM hemin plus 250 ng/mL Fe-MBT, respectively (Fig 2A). Three independent libraries were generated in Mtb ML1600 grown in media containing 250 ng/mL Fe-MBT and 20 μM hemin, respectively. Two independent libraries were generated for each of the other iron conditions. Thus, a total of 14 independent libraries, each containing approximately 200,000 transposon mutants, were obtained. Then, the chromosomal DNA was extracted from each pool, digested using the restriction enzyme *Hin*P1I, ligated to asymmetric adapters using a previously described method with modifications [39,40]. The transposon-chromosome junctions were amplified using PCR and subjected to Illumina sequencing (Fig 2B). An average of 8.5 million unique transposon-chromosome junction sequencing reads covered ~ 74% of the TA dinucleotide sites in the chromosome in each individual library (S5 Table), corresponding to approximately one insertion every 80 base pairs on average (Fig 2C). The high saturation of transposon insertions in our libraries also enabled us to assess the essentiality of the 1,312

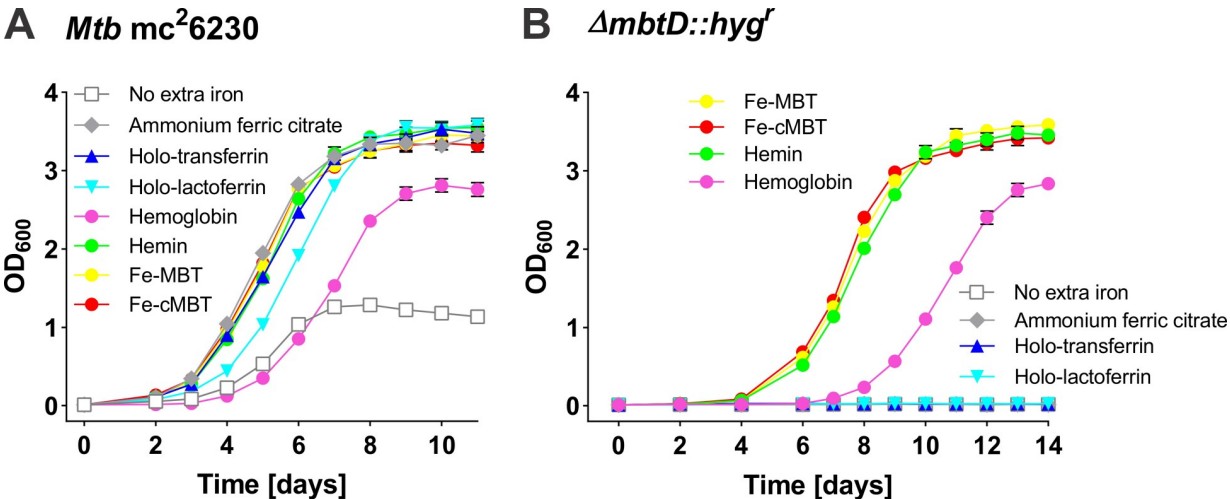

**Fig 1. Growth of wild-type *M. tuberculosis* and of the Δ*mbtD* mutant with different iron sources.** Growth of wild-type Mtb mc²6230 (A) and the Δ*mbtD::hyg^r* mutant (B) in self-made low-iron 7H9 medium supplemented with: 20 μM ammonium ferric citrate; 10 μM human holo-transferrin; 10 μM human holo-lactoferrin; 5 μM human hemoglobin; 20 μM hemin; 0.2 μM mycobactin (MBT) and 0.2 μM carboxymycobactin (cMBT), respectively. The Mtb strains were grown in self-made low-iron 7H9 medium for 5 days to deplete intracellular iron before growth analysis. The initial OD₆₀₀ of all the cultures was 0.01. Error bars represent standard deviations from the mean results of biological triplicates.

small ORFs ($\leq$ 9 TA sites), in contrast to earlier TnSeq studies of Mtb, in which small genes with fewer than 9 TA dinucleotide sites were not well-classified [39,41]. Only 75 to 90 small ORFs had no insertions. The classification of most of those genes (70–90) as essential across all iron conditions is similar to the results of a previous high density (84%) transposon library, which defined 92 small genes as essential after accounting for the presence of non-permissive TA sites [42].

### The genetic requirements for Mtb to utilize iron from siderophores and heme are drastically different

The intolerance of a gene to transposon (Tn) insertions reflects its importance under a given growth condition. This genetic requirement can be quantitatively assessed by the statistical analysis of the location and frequency of Tn insertions across a gene. Here, we used this quantitative information to determine the essentiality of genomic regions of Mtb when grown with different iron sources. First, we determined which genes exhibited statistically significant differences in Tn insertion counts across different iron sources using ANOVA. We identified 165 genes which exhibited a statistically significant number of insertions under at least one of the tested iron conditions, after correction for false discovery rates (adjusted $P$ values $<$ 0.05; S6 Table). The largest groups of varying genes were involved in intermediary metabolism and respiration, lipid metabolism, cell wall associated and conserved hypothetical proteins (Fig 3A). Hierarchical cluster analysis revealed iron utilization profiles that are specific for each of the six iron conditions (Fig 3B and S1 Fig). The genes required for utilization of hemin and hemoglobin as iron sources are very similar (Fig 3B, 3C and S1 Fig). This is consistent with our previous finding that hemoglobin and heme acquisition pathways overlap [36]. In the presence of hemin or hemoglobin as the sole iron source, Mtb exhibits increased requirements for genes involved in lipid metabolism, PDIM biosynthesis, ESX-3 secretion system, leucine biosynthesis, and decreased requirements for genes involved in heme/porphyrin biosynthesis, glycine metabolism, inorganic phosphate transporters, RND efflux pumps and Mce4 family proteins

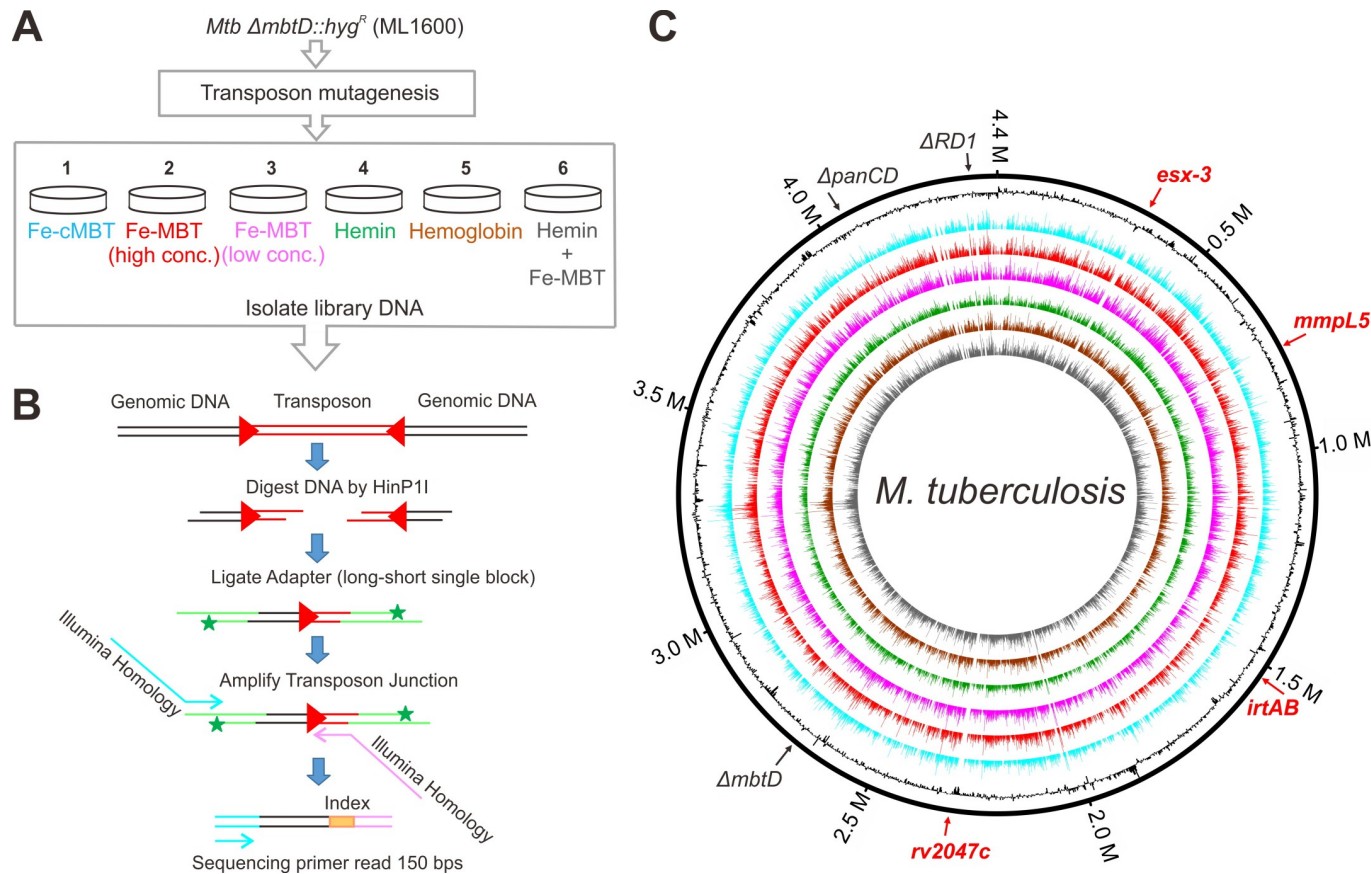

**Fig 2. Construction of *M. tuberculosis* transposon libraries.** (A) The Mtb transposon mutagenesis library was generated from the siderophore-deficient mutant ML1600 (Δ*mbtD::hyg^r^*). The library was cultured on low iron 7H9/ADS agar plates supplemented with (i) 500 ng/mL Fe-cMBT; (ii) 250 ng/mL Fe-MBT (high concentration); (iii) 50 ng/mL Fe-MBT (low concentration); (iv) 20 μM hemin; (v) 5 μM hemoglobin and (vi) 20 μM hemin plus 250 ng/mL Fe-MBT, respectively. The library DNA was isolated from the plates after three weeks. (B) Library building protocol. Genomic DNA pool was digested by *Hin*P1I to generate GC-tails, allowing to the ligation of GC-tailed adapters which compose of short 3′ blocked and long oligonucleotides. The 3′ end of the shorter oligo was blocked (*) by C6-TFA-amino modification so that it could not be extended during amplification. The transposon-chromosome junctions were selectively amplified using primers (S2 Table) that recognize the end of the transposon and the 5′ end of the long adapter oligo. These primers contain all requisite sequences to permit direct multiplex sequencing of amplicons on an Illumina NextSeq 500 platform. (C) Transposon-chromosome junctions from high density mutant libraries were identified by deep sequencing. The six concentric circles in different colors represent the six libraries corresponding to (A). From the outer to the inner, they are (i) cMBT; (ii) high MBT; (iii) low MBT; (iv) hemin; (v) hemoglobin and (vi) hemin plus high MBT, respectively. The number of normalized sequence reads corresponding to each insertion site is represented as one color bars mapped onto the circular chromosome of *M. tuberculosis* H37Rv. Black contour represents the GC content of the chromosome (G+C content higher or lower than 50% are represented as contours outside or inside the ring, respectively). The genomic deletion regions such as ΔRD1, Δ*panCD* and Δ*mbtD* are indicated in black. The genomic regions such as *esx-3*, *irtA/B*, *mmpL5* and *rv2047c* are indicated in red. Nucleotide positions are indicated (million bases). Plots were generated using Circos 0.69–5 [43].

(Fig 3B and S1 Fig). Interestingly, Mtb exhibits nearly opposite genetic requirements in the presence of siderophores as the sole iron source (Fig 3B, 3C and S1 Fig).

To further distinguish the iron utilization profiles in Mtb between siderophore and heme, we used a permutation test on insertion counts (resampling analysis in Transit [45]) and found that 64 and 54 genes are involved in utilization of MBT and heme, respectively, by comparing the number of transposon insertions in each gene in medium with high mycobactin concentrations (250 ng/mL) versus hemin (Fig 3D, S7 Table). In the presence of high MBT concentrations, Mtb exhibited significantly stronger requirements for genes encoding the RND efflux pumps MmpL5/MmpS5 and MmpL11, the heme/porphyrin biosynthesis enzymes (except HemD), the ABC transporters IrtA/IrtB, MurI (peptidoglycan biosynthesis), the pyruvate/phosphate dikinase PpdK, the polyketide synthase Pks12, MmaA4 (mycolic acid

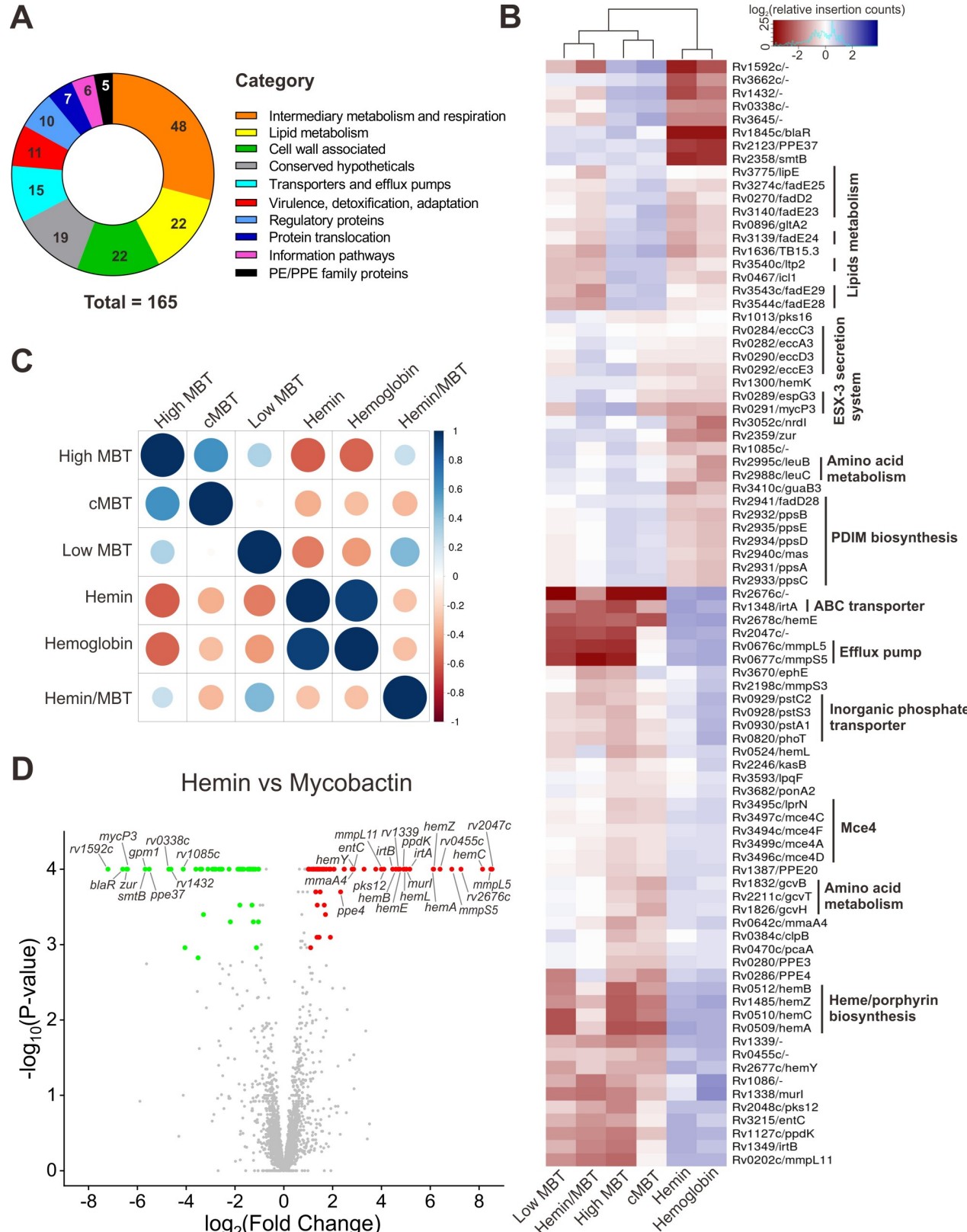

**Fig 3. TnSeq analysis of the *M. tuberculosis* transposon libraries.** A subset of 165 varying genes were identified by ANOVA. The mean normalized insertion count is calculated for each gene in each of the 6 conditions. Then a log-fold-change is calculated for each condition relative to the mean

count across all the conditions. (A) The 165 varying genes were categorized by different physiological functions as annotated in TubercuList [44]. (B) Heatmap with 84 selected genes (See full heatmap in S1 Fig) showing variability in transposon insertion counts across iron-supplementation conditions. The color 'red' means the counts in one condition are lower than the other conditions on average, suggesting a greater requirement for that gene in that condition, and 'blue' means insertion counts are higher than average, suggesting it is less required. The dendrogram shows the hierarchical clustering of the conditions (columns) using complete-linkage clustering. Gene insertion count profiles (rows) are also clustered using the *hclust* package in R, and gene pathway associations are indicated. (C) Correlation plot between each pair of iron-supplementation conditions. The correlation coefficients between pairs of conditions are calculated between vectors of log-fold-changes for the varying genes. The size of each circle indicates the magnitude of the correlation, and the color indicates the sign, as shown by the scale. (D) Volcano plot of resampling analysis between hemin and high MBT conditions. TnSeq-FC- and false discovery rate-adjusted *P* values (*q* values) from the resampling test are plotted for each genetic locus. Loci meeting the significance threshold of a *q* value of <0.05 are colored. The genes with increased log-fold-change or decreased minus log-fold-change are predicted to have more essentiality in utilizing mycobactin or hemin, respectively. Selected mutants are indicated by name.

modification) and proteins of unknown functions such as Rv2047c and Rv0455c (Fig 3D, S7 Table). By contrast, in the presence of hemin, Mtb had significantly stronger requirements for the putative secretory lipase Rv1592c, the sensor-transducer protein BlaR, the zinc-responsive transcriptional repressors Zur and SmtB, PPE37, the dehydrogenase Rv1432, the iron-sulfur-binding reductase Rv0338c, the ESX-3 component MycP3, the hemolysin-like protein Rv1085c, Gpm1 (carbon and amino acids metabolism), and the electron transfer flavoprotein FixAB (Fig 3D, S7 Table). Taken together, our TnSeq experiments revealed the distinct iron-source dependent genetic requirements by Mtb.

## Siderophore utilization by *M. tuberculosis*

To classify the contribution of Mtb genes to the utilization of different iron sources by Mtb, we used a Hidden Markov Model (HMM) which defines four different states of essentiality: essential (ES), growth defect (GD), non-essential (NE), and growth advantage (GA) [42]. Our study identified 428 genes which are essential or required for optimal growth (GD) of Mtb and shared among the five conditions with a single iron source (Fig 4), suggesting that these genes are generally required by Mtb and not involved in iron-specific processes. Indeed, almost all of these shared genes (96.5%) were previously identified as essential or required for optimal growth by Mtb in enriched Middlebrook medium replete with iron salts [42].

The ESX-3 secretion system was previously shown to be required for siderophore-mediated iron acquisition in mycobacteria [17,19]. As anticipated, most genes of the *esx-3* operon were classified by HMM analysis as 'essential' or 'growth-defect' for Mtb growing with cMBT (500 ng/mL) or low concentrations of MBT (50 ng/mL) (S8 Table, Fig 5A). Interestingly, significantly more insertions were observed in the *ppe4* gene when Mtb was grown with high concentrations of MBT (250 ng/mL) (Fig 5A and S8 Table). Thus, *ppe4* was classified as non-essential in medium with high MBT, in contrast to Mtb grown with cMBT (500 ng/mL) or low concentrations of MBT (50 ng/mL). These results are consistent with previous experiments which show that neither the Δ*esx-3* mutant nor the Δ*pe5-ppe4* mutant grew on 7H10 plates with low concentrations of MBT (2 to 20 ng/mL), but that they were rescued by high concentrations of MBT (200 ng/mL) [19]. Surprisingly, the *esx-3* operon was also assessed to be essential for Mtb grown with hemin or hemoglobin as the sole iron source (Fig 5A, S8 Table). No insertions were observed in any of the *esx-3* genes except for *ppe4* (Fig 5A). The *ppe4* gene tolerated insertions in the 3' end when Mtb was grown with heme or hemoglobin, while the 5' end of *ppe4* appeared to be essential for Mtb under all conditions (Fig 5A, Table 1). These findings are consistent with previous observations that hemin does not support growth of either the Δ*esx-3* or the Δ*pe5-ppe4* mutant on 7H10 plates, unless 0.05% Tween 80 was included [19]. In addition, *ppe3* and *ppe20* mutants had significantly reduced fitness at high concentrations of MBT or with cMBT (Fig 3B, S7 Table). In conclusion, these results indicated that PPE4 is the most important PPE protein of Mtb for iron utilization.

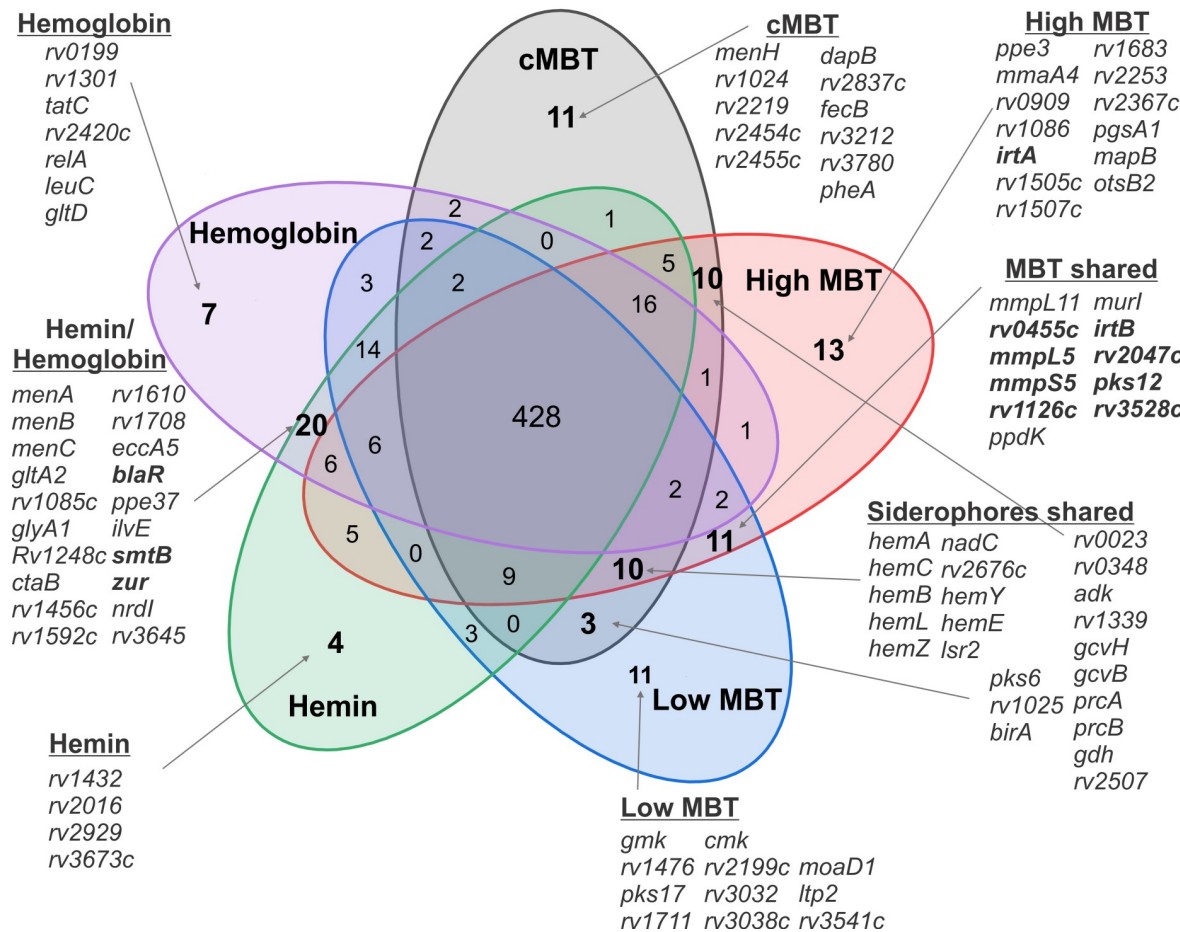

**Hemoglobin**
rv0199
rv1301
tatC
rv2420c
relA
leuC
gltD

**Hemin/Hemoglobin**
| | |
|---|---|
| menA | rv1610 |
| menB | rv1708 |
| menC | eccA5 |
| gltA2 | **blaR** |
| rv1085c | ppe37 |
| glyA1 | ilvE |
| Rv1248c | **smtB** |
| ctaB | **zur** |
| rv1456c | nrdI |
| rv1592c | rv3645 |

**Hemin**
rv1432
rv2016
rv2929
rv3673c

**cMBT**
| | |
|---|---|
| menH | dapB |
| rv1024 | rv2837c |
| rv2219 | fecB |
| rv2454c | rv3212 |
| rv2455c | rv3780 |
| | pheA |

**High MBT**
| | |
|---|---|
| ppe3 | rv1683 |
| mmaA4 | rv2253 |
| rv0909 | rv2367c |
| rv1086 | pgsA1 |
| **irtA** | mapB |
| rv1505c | otsB2 |
| rv1507c | |

**MBT shared**
| | |
|---|---|
| mmpL11 | murI |
| **rv0455c** | **irtB** |
| **mmpL5** | **rv2047c** |
| **mmpS5** | **pks12** |
| **rv1126c** | **rv3528c** |
| ppdK | |

**Siderophores shared**
| | | |
|---|---|---|
| hemA | nadC | rv0023 |
| hemC | rv2676c | rv0348 |
| hemB | hemY | adk |
| hemL | hemE | rv1339 |
| hemZ | lsr2 | gcvH |
| | pks6 | gcvB |
| | rv1025 | prcA |
| | birA | prcB |
| | | gdh |
| | | rv2507 |

**Low MBT**
| | | |
|---|---|---|
| gmk | cmk | |
| rv1476 | rv2199c | moaD1 |
| pks17 | rv3032 | ltp2 |
| rv1711 | rv3038c | rv3541c |

**Fig 4. Quantitative classification of the requirement of genes by *M. tuberculosis* for growth under different iron conditions.** Venn diagram of Mtb genes with 'essential' or 'growth defect' states from five iron conditions: cMBT (500 ng/mL), high MBT (250 ng/mL), low MBT (50 ng/mL), hemin (Hm) (20 μM) and hemoglobin (Hb) (5 μM). The number of overlapping genes in their respective zones is indicated.

## Siderophore poisoning of Mtb

MmpS4 and MmpS5 are the only proteins known to be required for siderophore secretion by Mtb [12]. Our TnSeq analysis indicated that *mmpS4* (*rv0451c*, 15 TA sites) was essential under all conditions containing single iron sources (S3A Fig, S8 Table). The important role of MmpS4 is consistent with our previous finding that MmpS4 can substitute for MmpS5 in siderophore secretion but not vice versa [12], indicating that *mmpS4* is essential for siderophore secretion under these conditions. However, the Mtb *mmpS4* deletion mutant did not show a growth defect *in vitro* [12]. These divergent results might reflect the differences between experiments with pooled transposon mutants in the TnSeq experiments and isogenic mutants grown individually [12]. The essentiality for MmpS4 by Mtb grown with hemin and hemoglobin as iron sources is unexpected (S3A Fig, S8 Table) and indicates an unknown metabolic role of MmpS4 in addition to siderophore secretion.

Our TnSeq analysis also showed no transposon insertions in *mmpS5* (*rv0677c*, 6 TA sites) in the presence of 250 ng/mL MBT ("high") (Fig 5B, S8 Table). In contrast to *mmpS4*, *mmpS5* is not essential for Mtb grown with heme or hemoglobin as sole iron sources (S3A Fig, S8 Table). The *mmpL5* gene (*rv0676c*, 32 TA sites) is located in the same operon as *mmpS5* and

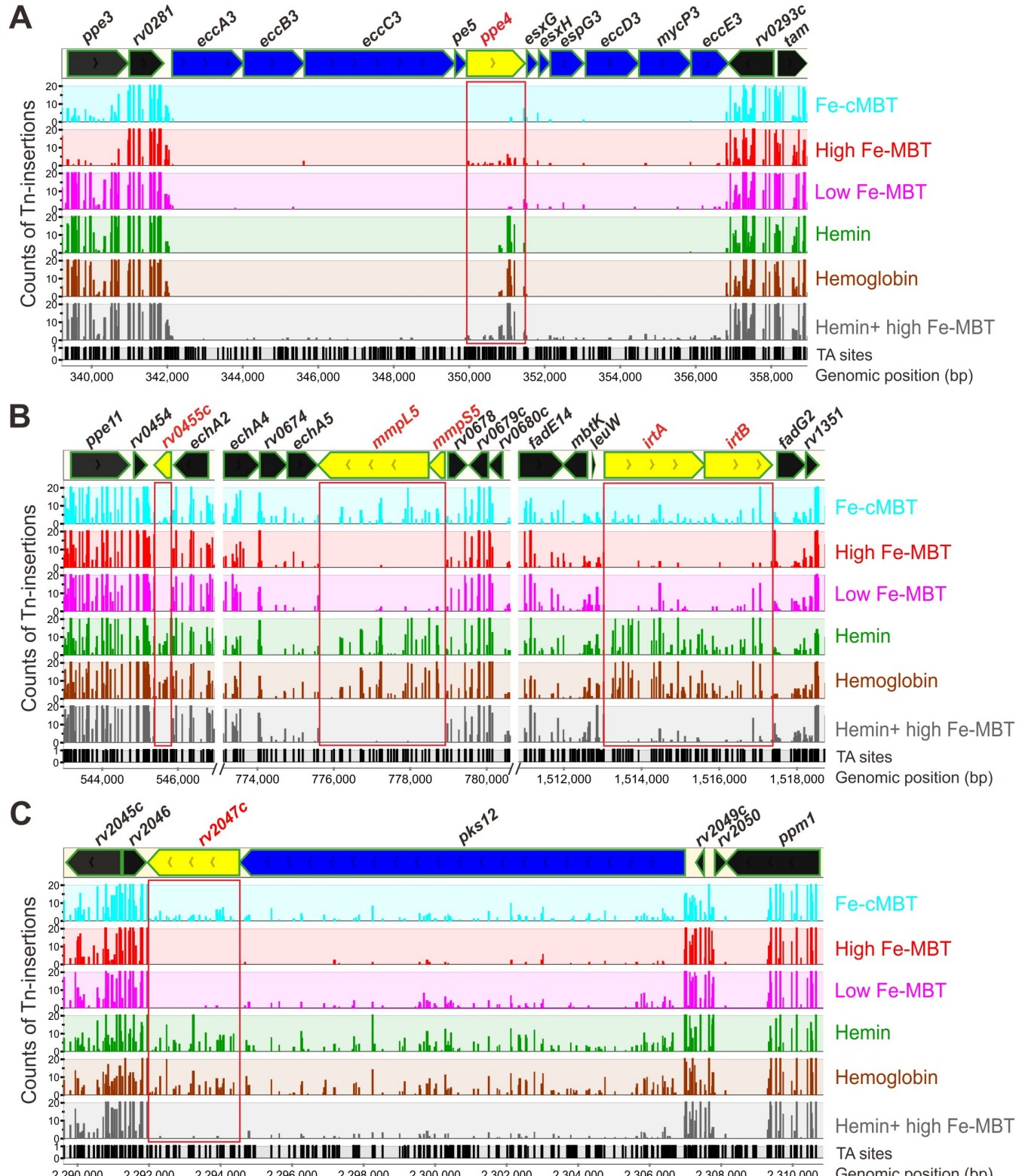

**Fig 5. *M. tuberculosis* genomic regions involved in siderophore utilization.** The distribution of the transposon insertions in Mtb genomic regions including *esx-3* operon (A); *rv0455c* gene, *mmpS5-mmpL5* operon, *irtA-irtB* operon (B); and *pks12-rv2047c* operon (C). The iron conditions are marked with different colors. The y-axis (0, 10, 20) represents the counts of the Tn-insertions and x-axis represents the genomic position (bp). Potential TA dinucleotide insertions sites are indicated. Regions containing genes of interest are highlighted by red boxes. Plots were generated using MochiView 1.46 [46].

**Table 1. Genes with differential importance in iron utilization by *Mycobacterium tuberculosis*.** 45 genes selected from the 165 varying genes which were assessed by HMM analysis as either essential (ES), "essential domain" (ESD) or "growth defect" (GD) under different iron conditions. The 21 genes which were classified as essential or "growth defect" under all iron conditions (S6 Table) were excluded from this list. cMBT: carboxymycobactin, MBT: mycobactin at high or low concentrations, hemin (Hm), hemoglobin (Hb) and hemin plus mycobactin (Hm/MBT). NE: non-essential.

| ORF | Gene | Description | cMBT | MBT (high) | MBT (low) | Hm | Hb | Hm/ MBT |
|---|---|---|---|---|---|---|---|---|
| rv0202c | mmpL11 | RND efflux pump | NE | GD | GD | NE | NE | GD |
| rv0280 | ppe3 | PPE protein | NE | GD | NE | NE | NE | NE |
| rv0286 | ppe4 | PPE protein and ESX-3 substrate | ES | NE | ES | ESD | ESD | NE |
| rv0338c | | Iron-sulfur-binding reductase | NE | NE | ES | ES | ES | GD |
| rv0455c | | Hypothetical protein | NE | ES | ES | NE | NE | ES |
| rv0509 | hemA | Glutamyl-tRNA reductase | ES | ES | GD | NE | NE | NE |
| rv0510 | hemC | Porphobilinogen deaminase | GD | ES | GD | NE | NE | NE |
| rv0512 | hemB | Delta-aminolevulinic acid dehydratase | ES | ES | GD | NE | NE | NE |
| rv0524 | hemL | Glutamate-1-semialdehyde aminotransferase | GD | ES | GD | NE | NE | NE |
| rv0558 | menH | Demethylmenaquinone methyltransferase | GD | NE | NE | NE | NE | GD |
| rv0642c | mmaA4 | Hydroxymycolate synthase | NE | GD | NE | NE | NE | NE |
| rv0676c | mmpL5 | RND efflux pump for siderophore secretion | NE | ES | GD | NE | NE | GD |
| rv0677c | mmpS5 | Mycobacterial membrane protein small | NE | ES | GD | NE | NE | GD |
| rv0896 | gltA2 | Citrate synthase 1 | NE | NE | NE | GD | GD | NE |
| rv1085c | | Annotated hemolysin-like protein | NE | NE | NE | GD | GD | NE |
| rv1086 | | Short-chain Z-isoprenyl diphosphate synthase | NE | GD | NE | NE | NE | GD |
| rv1127c | ppdK | Pyruvate phosphate dikinase | NE | ES | GD | NE | NE | ES |
| rv1248c | | Multifunctional 2-oxoglutarate dehydrogenase | NE | NE | NE | GD | GD | GD |
| rv1300 | hemK | Release factor glutamine methyltransferase | GD | NE | NE | NE | GD | NE |
| rv1338 | murI | Glutamate racemase | NE | GD | GD | NE | NE | GD |
| rv1339 | | Beta-lactamase superfamily | GD | GD | NE | NE | NE | GD |
| rv1348c | irtA | ABC transporter for siderophore uptake | NE | GD | NE | NE | NE | NE |
| rv1349c | irtB | ABC transporter for siderophore uptake | NE | GD | GD | NE | NE | GD |
| rv1432 | | Dehydrogenase | NE | NE | NE | GD | NE | NE |
| rv1485 | hemZ | Ferrochelatase | ES | ES | ES | NE | NE | NE |
| rv1592 | | Putative secretory lipase | NE | NE | NE | GD | GD | NE |
| rv1826 | gcvH | Glycine cleavage system protein H | ES | ES | NE | NE | NE | NE |
| rv1832 | gcvB | Glycine dehydrogenase | ES | GD | NE | NE | NE | NE |
| rv1845c | blaR | Sensor-transducer protein | NE | NE | NE | ES | ES | NE |
| rv2047c | | Hypothetical protein with PEP-utilizing domain | NE | ES | ES | NE | NE | GD |
| rv2048c | pks12 | Polyketide synthase Pks12 | NE | GD | GD | NE | NE | GD |
| rv2123 | ppe37 | PPE protein | NE | NE | NE | GD | GD | NE |
| rv2221c | glnE | Glutamine-synthetase adenylyltransferase | NE | GD | NE | ES | ES | GD |
| rv2358 | smtB | Zinc-responsive transcriptional repressor | NE | NE | NE | ES | ES | NE |
| rv2359 | zur | Zinc uptake regulation protein | NE | NE | NE | ES | ES | NE |
| rv2454c | | 2-oxoglutarate oxidoreductase subunit | GD | NE | NE | NE | NE | NE |
| rv2676c | hemQ | Chlorite dismutase | ES | ES | GD | NE | NE | NE |
| rv2677c | hemY | Protoporphyrinogen oxidase | ES | GD | ES | NE | NE | NE |
| rv2678c | hemE | Uroporphyrinogen decarboxylase | ES | GD | ES | NE | NE | ES |
| rv2988c | leuC | 3-isopropylmalate dehydratase large subunit | NE | NE | NE | NE | GD | NE |
| rv3052c | nrdI | Protein involved in ribonucleotide reduction | NE | NE | NE | ES | ES | NE |
| rv3267 | | Hypothetical protein | GD | GD | NE | GD | NE | NE |
| rv3543c | fadE29 | acyl-CoA dehydrogenase | NE | NE | NE | NE | NE | ES |
| rv3645c | | Membrane protein with adenylate cyclase domain | NE | NE | NE | GD | GD | NE |
| rv3859c | gltB | Glutamate synthase large subunit | NE | NE | GD | NE | GD | GD |

the encoded proteins were proposed to interact [12]. Indeed, the TnSeq results for *mmpL5* mimicked the results for *mmpS5*: In the presence of 250 ng/mL MBT the *mmpL5* gene had almost no transposon insertions, but most of the 32 TA sites had insertions in the presence of heme and hemoglobin as sole iron sources (Fig 5B, S8 Table). Importantly, hemin did not rescue the Mtb mutants with Tn insertions in *mmpS5* or *mmpL5* in the presence of 250 ng/mL MBT, in contrast to genes involved in siderophore uptake such as *ppe4* (Fig 5A). This is likely due to the previously observed toxicity of mycobactin and carboxymycobactin for Mtb mutants defective in siderophore secretion regardless of the presence of other iron sources in the medium [13]. Interestingly, the *irtA* and *irtB* genes encoding the siderophore importer IrtAB showed a similar pattern of transposon insertions (Fig 5B). *IrtA* and *irtB* were classified as 'growth defect', but not as essential, at high concentrations of MBT (Table 1), indicating that there might be alternative transporters for siderophore uptake. This conclusion is consistent with the previous growth assays in which *irtA* and *irtB* mutants had growth defects in iron-deficient medium but not in iron-replete medium [10]. Although *irtA* and *irtB* were found to be not essential in the presence of hemin or hemoglobin, adding high concentrations of MBT in the presence of hemin significantly reduced the fitness of Mtb mutants with insertions in *irtA* and *irtB* indicating that blocking siderophore uptake across the inner membrane is also toxic for Mtb, possibly by increased siderophore accumulation in the periplasm (Figs 3B and 5B, S7 Table).

We identified two other genes which showed the siderophore poisoning phenotype: *rv0455c* and *rv2047c* (Fig 5B and 5C). To examine experimentally whether siderophores are indeed toxic for Mtb mutants lacking these genes TnSeq analysis is indeed capable of identifying genes with a similar phenotype as *mmpS5* and *mmpL5*, we constructed an isogenic *rv2047c* deletion mutant in Mtb mc²6230 (H37Rv ΔRD1 ΔpanCD) using the vector pML3605 (S4 Table). While the parent strain Mtb mc²6230 and the Δ*rv2047c* deletion mutant showed similar growth rates in 7H9 medium (without ADS) containing 0.01% Tyloxapol, the addition of 1 μM MBT completely inhibited growth of the Δ*rv2047c* mutant (Fig 6). The growth of the Δ*rv2047c* mutant was only modestly reduced in the presence of 1 μM cMBT. This observation is consistent with our previous finding that MBT is much more toxic for siderophore secretion-defective mutants compared to cMBT [13]. Growth of the Δ*rv2047c* mutant was fully restored by expressing the *rv2047c* gene using a chromosomal expression vector pML4211 integrated at the L5 *attB* site (S4 Table). These results demonstrated that *rv2047c* is essential for Mtb to grow in the presence of exogenous high concentrations of MBT, further validating our TnSeq analysis.

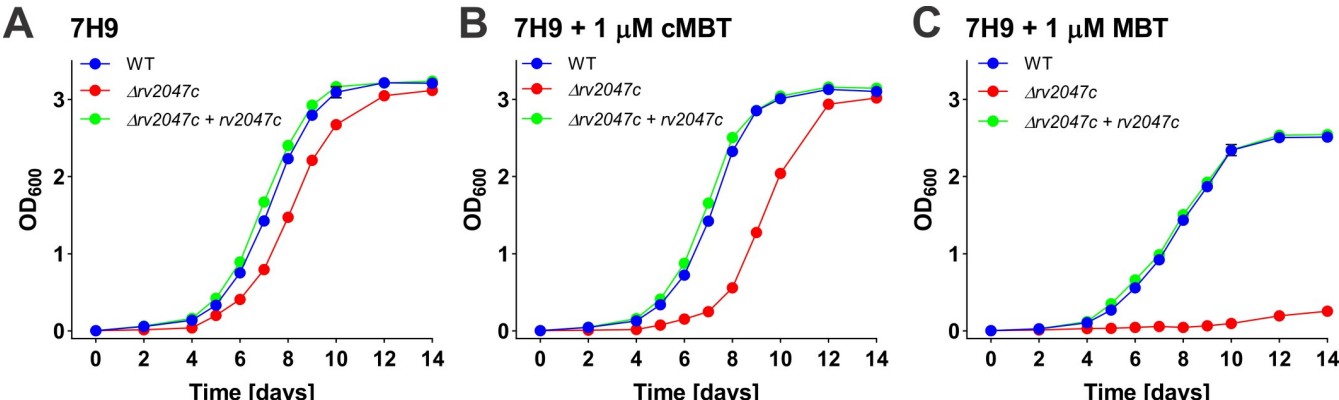

**Fig 6. Growth of the *M. tuberculosis* Δ*rv2047c* mutant.** Growth of Mtb mc²6230 (as WT), Δ*rv2047c* (ML2256) and the complementation Δ*rv2047c* mutant (ML2257) in 7H9 Middlebrook medium (A) supplemented with 1 μM carboxymycobactin (B) and 1 μM mycobactin (C), respectively. The initial OD$_{600}$ of all the cultures is 0.01. Error bars represent standard deviations from the mean results of biological triplicates.

## Roles of the MmpL4/MmpS4 and MmpL5/MmpS5 efflux systems in siderophore-mediated iron utilization by *M. tuberculosis*

The small membrane proteins MmpS are encoded in the same operon as their associated large transmembrane proteins MmpL [47]. This association, and the interaction of the MmpS4 protein with MmpL4 [12], has led to the assumption that the MmpS and their MmpL counterparts are involved in the same functions [48]. This appears to be true for *mmpS5* and *mmpL5*, as Mtb has identical requirements for these genes for growth on different iron sources (Fig 3B and 3D, S7 Table). Surprisingly, *mmpL4* was not required by Mtb for utilization of any iron source, in stark contrast to *mmpS4*, which was essential under all conditions containing a single iron source (S3A Fig, S8 Table). To further examine the roles of the *mmpL4* and *mmpL5* genes in iron utilization by Mtb, we determined the growth of the isogenic Mtb mutants lacking either one or both of these genes (S1 Table). The Δ*mmpL4/L5* double mutant (Mtb ML2302) did not grow at all in low-iron 7H9 medium, while both single mutants, Mtb ML2300 (Δ*mmpL4*) and Mtb ML2301 (Δ*mmpL5*) grew like the parent Mtb strain (Fig 7A). As expected, the mycobactin biosynthesis mutant Mtb Δ*mbtD* [23] and the siderophore secretion-defective mutant Mtb Δ*mmpS4/S5* [12] did not grow in low-iron 7H9 medium (Fig 7A). All mutants with a growth defect in low-iron 7H9 medium were rescued at least partially with 10 μM hemin (Fig 7B). By contrast, adding 1 μM cMBT and 1 μM MBT as the sole iron sources rescued only the mycobactin biosynthesis mutant Δ*mbtD*, as expected [23], while growth of the Δ*mmpS4/S5* and the Δ*mmpL4/L5* double mutants was completely suppressed (Fig 7C and 7D). The phenotypes of the double mutants are identical, indicating that the *mmpL4* and *mmpL5* genes are involved in siderophore secretion as described for *mmpS4* and *mmpS5* [12]. Thus, the complete growth inhibition of the Δ*mmpL4/L5* double mutant is likely due to self-poisoning by intracellular accumulation of siderophores, as shown for the Mtb Δ*mmpS4/S5* mutant [13]. Taken together, these results indicated that MmpL4, together with MmpL5, is required for siderophore secretion in Mtb, and that MmpL5 plays a major role in this process.

## Heme and hemoglobin utilization by *M. tuberculosis*

The nearly opposite requirement for genes depending on whether Mtb utilized iron from heme or hemoglobin versus siderophore (Fig 3B and 3D) indicated that there is little, if any, overlap between these iron acquisition pathways. Transposon insertions in 31 genes were uniquely required by Mtb to grow with hemin and hemoglobin as iron sources, but not with siderophores (Fig 4, S9 Table). Among those genes, we found that Mtb exhibits significantly stronger requirements for *ppe37* and *rv1085c* encoding a hemolysin-like protein (Figs 3D and 8A; S7 and S9 Tables). The growth defect of Mtb mutants with insertions in *ppe37* is consistent with previous findings that PPE37 is involved in heme utilization in the Mtb Erdman strain [27]. However, the requirement for Rv1085c for growth of Mtb *in vitro* in the absence of erythrocytes indicates that the molecular function of Rv1085c is probably different from that of hemolysins [49]. The TnSeq analysis also identified *blaR* (*rv1845c*), *zur* and *smtB* as essential for Mtb to utilize heme or hemoglobin as iron sources (Figs 3D and 8A; S7 and S9 Table, Table 1), indicating that gene activation is a necessary component in this process. To examine this question experimentally, we focused on SmtB and Zur, two transcriptional regulators, which are encoded in the same operon and have been shown to be involved in zinc uptake [50]. To this end we deleted the *smtB-zur* operon in the avirulent Mtb mc²6206 strain (Δ*panCD* Δ*leuCD*) to obtain Mtb ML2277 (Δ*smtB-zur*) (S1 Table, S5 Fig). Mtb ML2277 was complemented with chromosomal expression vectors for the *zur* and *smtB* genes and for the *smtB-zur* operon, respectively. Integration of these vectors into the L5 *attB* site created the Δ*zur* single mutant ML2278 (ML2277::*smtB*), the Δ*smtB* single mutant ML2279 (ML2277::

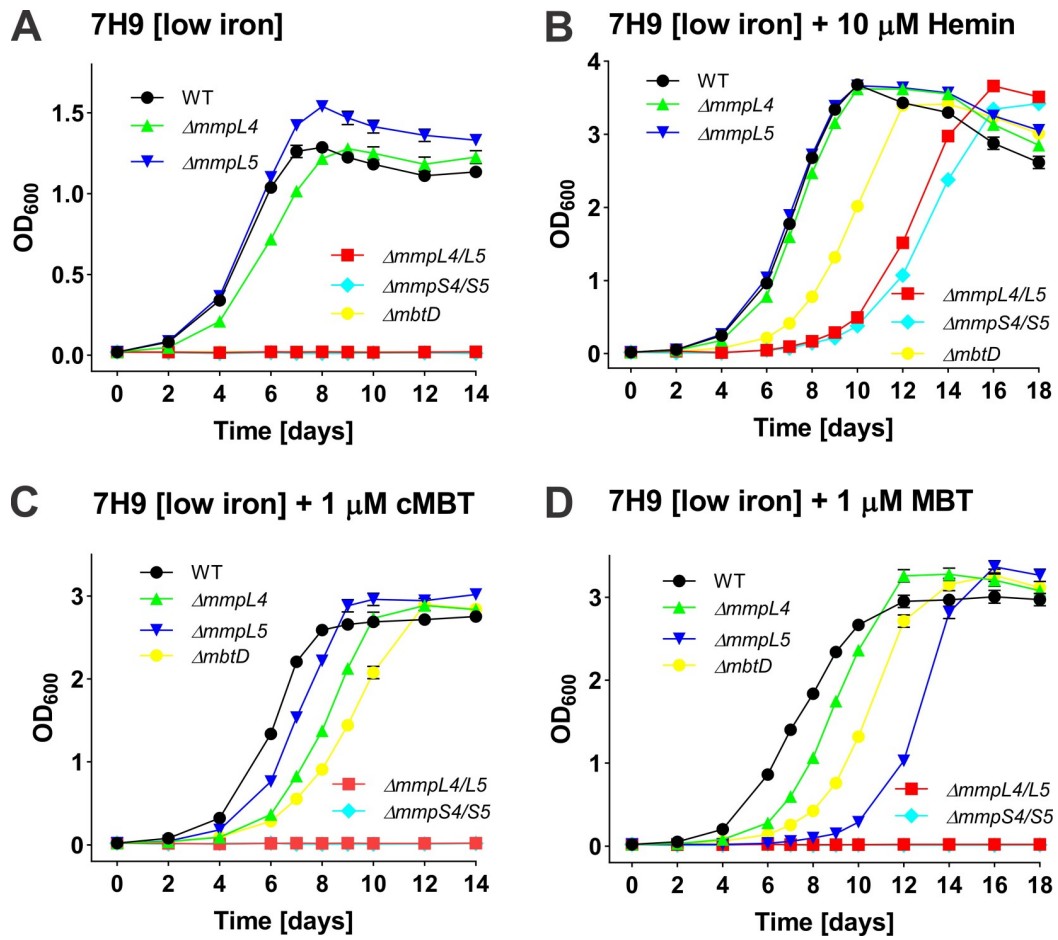

**Fig 7. Growth of the *M. tuberculosis* *mmpL4* and *mmpL5* mutants under different iron conditions.** Growth curves of Mtb strains including Mtb mc$^2$6230 (as WT), Δ*mmpL4* (ML2300), Δ*mmpL5* (ML2301), Δ*mmpL4/L5* (ML2302), Δ*mmpS4/S5* (ML859) and Δ*mbtD* (ML1600) in low-iron 7H9 medium (A) supplemented with 10 μM hemin (B), 1 μM carboxymycobactin (C) and 1 μM mycobactin (D), respectively. The Mtb strains were grown in self-made low-iron 7H9 medium for 5–7 days to deplete intracellular iron before growth assays. The initial OD$_{600}$ of all the cultures is 0.01. Error bars represent standard deviations from the mean results of biological triplicates.

*zur*), and the fully complemented strain ML2280 (ML2277::*smtB-zur*) (S5 Fig, S1 Table). The parent Mtb mc$^2$6206 strain, the fully complemented double mutant and the Mtb strain lacking only *zur* had no or only a minor growth defect in low iron or replete iron 7H9 medium, while the strain lacking *smtB* and the Δ*smtB-zur* double mutant had a general growth defect independent of the iron concentration as shown in Figs 8B and S6. Collectively, these results suggested that SmtB and Zur are not required for utilization of Fe$^{3+}$ ions. Growth of the strain lacking only *zur* was significantly reduced in the presence of 20 μM hemin as the sole iron source in contrast to iron-replete or iron-deficient conditions, while the strain lacking SmtB had a similar growth defect compared to the parent strain in media with low iron and with hemin (Fig 8B and 8C and S6 Fig). These results indicated that Mtb requires Zur but not SmtB for hemin utilization. No growth of the Δ*smtB-zur* double mutant was observed in medium with hemoglobin as the sole iron sources (Fig 8D). Interestingly, only expression of both *smtB* and *zur* enhanced growth of the Δ*smtB-zur* double mutant, indicating that both regulators are essential for Mtb to grow with hemoglobin as the sole iron source (Fig 8D).

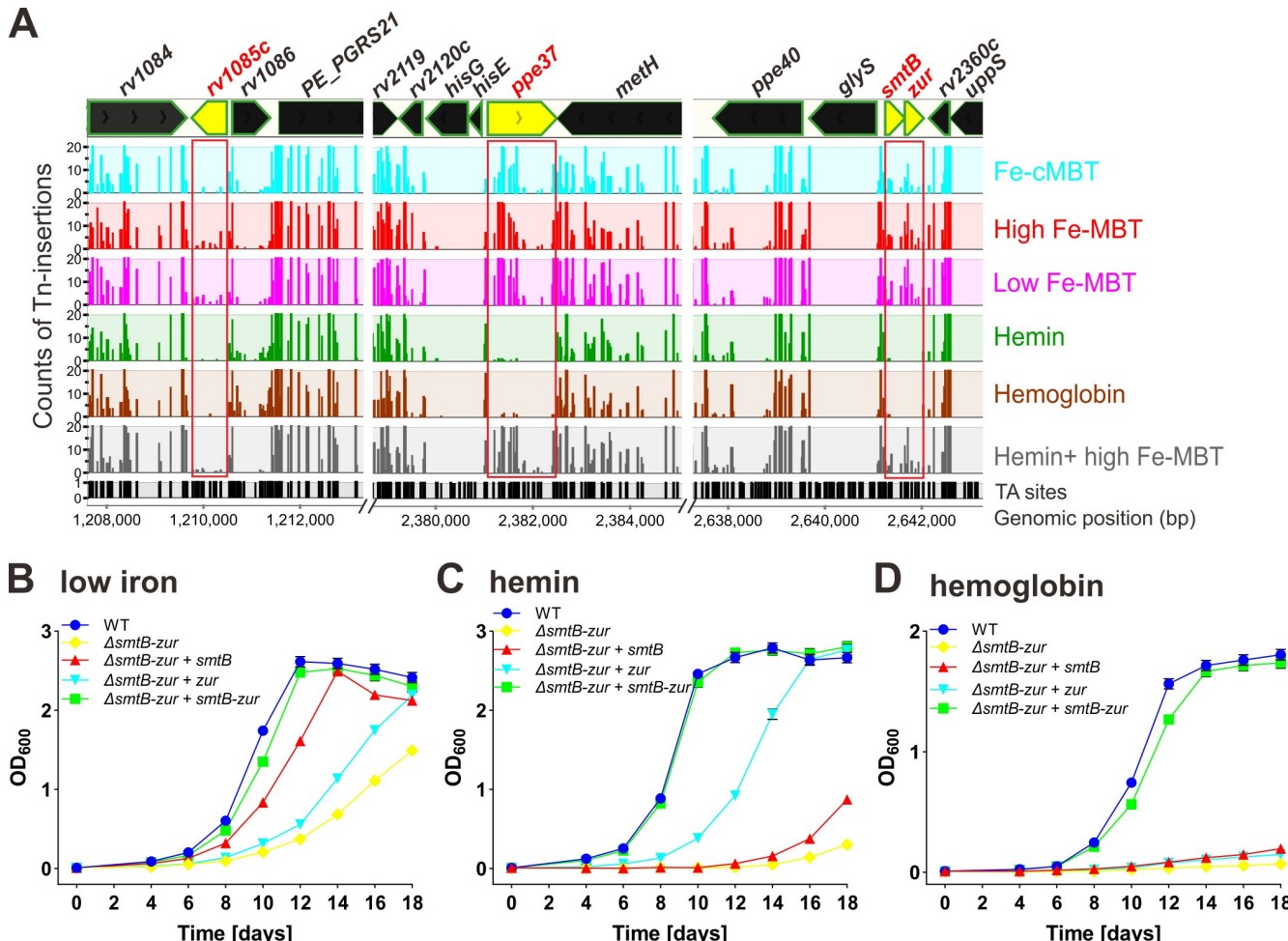

**Fig 8. *M. tuberculosis* genes involved in heme and hemoglobin utilization.** (A) The distribution of the transposon insertions in Mtb genomic regions including *rv1085c*, *ppe37* and *smtB-zur* operon. The iron conditions are marked with different colors. The y-axis (0, 10, 20) represents the counts of the Tn-insertions and x-axis represents the genomic position (bp). Potential TA dinucleotide insertions sites are indicated. Regions containing genes of interest are highlighted by red boxes. Plots were generated using MochiView. (B-D) Growth assays of wild-type Mtb mc²6206, Δ*smtB-zur* deletion mutant (ML2277), *smtB* complementation strain (ML2278), *zur* complementation strain (ML2279), and *smtB-zur* complementation strain (ML2280) under low-iron 7H9 medium containing 10 μM 2,2'-dipyridyl (DIP) (B) supplemented with 20 μM hemin (C) and 5 μM hemoglobin (D), respectively. The Mtb strains were grown in self-made low-iron 7H9 medium for 7 days to deplete intracellular iron before growth assays. The initial $OD_{600}$ of all the cultures is 0.01. Error bars represent standard deviations from the mean results of biological triplicates.

## Discussion

In this study, we used transposon insertion libraries generated in a siderophore-deficient Mtb strain and deep sequencing to assess the essentiality of genes for utilization of different iron sources by Mtb without the confounding effects of internal siderophore production. Transposon inactivation of 69 genes resulted in a significant or complete growth defect of Mtb only in the presence of siderophores, while 31 genes were uniquely required in the presence of hemin and/or hemoglobin (Fig 4, S9 Table).

### Siderophore utilization

Among the 69 genes predicted to be required for Mtb for cMBT and/or MBT utilization were *irtA*, *irtB* and *mmpS5* which were previously shown to be involved in carboxymycobactin uptake [10] and siderophore secretion [12] validating the TnSeq analysis. In a previous TnSeq

study using a saturated transposon library, 27 of these genes (S9 Table) were identified as essential for growth of Mtb in rich 7H9 or 7H10 Middlebrook media supplemented with oleic acid, albumin, dextrose and catalase [42]. Interestingly, none of these genes are required when Mtb utilizes heme or hemoglobin as an iron source. These genes encode proteins with functions in intermediary metabolism such as NadC (NAD$^+$ biosynthesis) or OtsB2 (trehalose metabolism), in protein degradation such as PrcA and PrcB (proteasome) or in cell wall biosynthesis such as MurI (glutamate racemase) (S9 Table), indicating that the iron source has a surprisingly strong influence on Mtb physiology well beyond the proteins directly involved in acquisition of these iron sources. One example is the *hem* genes which are required for porphyrin biosynthesis and were only found to be essential when siderophores were used as iron sources (Fig 4, S8 Table). This is consistent with previous TnSeq studies which used standard Mtb growth medium containing iron salts which do not contain heme [39,42,51] because Mtb cannot utilize iron salts without siderophores [13]. Importantly, the porphyrin biosynthesis genes are not essential when Mtb grows with hemin as the sole iron source indicating that Mtb is capable of utilizing protoporphyrin IX after iron is removed from heme and that genes involved in porphyrin biosynthesis are conditionally essential for Mtb *in vitro* growth.

## The siderophore-dependent self-poisoning phenotype

An interesting observation was that a few Mtb genes such as *mmpS5*, *mmpL5*, *rv2047c* and *rv0455c* were essential for growth in medium with high concentrations of MBT in the presence of hemin, although the same genes were not required for growth in medium with hemin or hemoglobin alone. This phenotype cannot be explained by the dependency of Mtb growth on a particular iron source since hemin can rescue mutants with defects in siderophore utilization and vice versa [23,26,36]. This phenotype, however, mimics the self-poisoning phenotype observed for the ΔmmpS4/ΔmmpS5 siderophore secretion mutant [13] and for the ΔmmpL4/ ΔmmpL5 in this study. Indeed, growth experiments confirmed that MBT is toxic for an isogenic *rv2047c* deletion mutant of Mtb, demonstrating that TnSeq experiments are a valuable tool to identify the siderophore-dependent self-poisoning phenotype in Mtb. The *rv2047c* gene encodes a protein with two predicted domains, an N-terminal NAD$^+$-dependent epimerase domain and a C-terminal phosphoenolpyruvate-utilizing domain. The *rv2047c* gene is located in an operon with *pks12* which encodes a polyketide synthetase required for the biosynthesis of mannosyl-β-1-phosphomycoketides, also called mycoketides or MPMs [52]. Interestingly, Mtb also exhibited a strong requirement for *pks12* in the presence of MBT (Figs 3B, 3D and 4) indicating that Rv2047c might be involved in mycoketide biosynthesis. However, it is unclear whether the role of Rv2047c in the siderophore-dependent self-poisoning phenotype is direct or indirect. It is possible that disruption of mycoketide biosynthesis compromises the outer membrane permeability barrier and enables uncontrolled and possibly poisonous influx of MBT into Mtb. Alternatively, mycoketides might be required for the function of the siderophore secretion system.

Transposon insertions into the *rv0455c* gene also showed a pattern consistent with siderophore poisoning (Fig 5B). The encoded Rv0455c protein contains a predicted N-terminal Sec-secretion peptide and was found in Mtb culture filtrates [53]. It is possible that Rv0455c transports MBT to the extracellular medium in Mtb. However, an indirect role of Rv0455c in siderophore secretion cannot be excluded. Thus, further genetic and biochemical experiments are needed to reveal the molecular function of Rv0455c.

## Roles of the MmpL4 and MmpL5 efflux pumps in siderophore secretion

The *mmpL5* gene showed a concentration-dependent essentiality in the TnSeq experiments: It was classified as "growth defect" in the presence of low concentrations of MBT (50 ng/mL)

and as essential in the presence of high concentrations of MBT (250 ng/mL) (Fig 3B and 3D; S8 Table). The isogenic Δ*mmpL5* deletion mutant showed a growth defect in liquid medium with 1 μM MBT as the sole iron source (Fig 7D), but no growth defect with 1 μM cMBT (Fig 7C). The stronger phenotype of Mtb siderophore secretion mutants in the TnSeq experiments in the presence of MBT versus cMBT is consistent with the six-fold increased toxicity of MBT over cMBT for the Δ*mmpS4/S5* mutant [13]. This might be due to the increased toxicity of the more hydrophobic MBTs due to their membrane association. This phenotype of *mmpL5* mutants is in contrast to the *mmpL4* gene which was classified as non-essential in the TnSeq experiments. This classification was confirmed by the absence of any growth defect of the isogenic Δ*mmpL4* mutant (Fig 7D). The increased importance of *mmpL5* compared to *mmpL4* is consistent with the approximately 7-fold higher transcription level of *mmpL5* during *in vitro* growth of Mtb [54] and the importance of MmpL5 in efflux of bedaquline, clofazimine and azoles compared to other MmpL proteins [55,56,57]. It is important to note that some of the results obtained by TnSeq are in apparent contradiction with results from growth experiments with isogenic Mtb mutants. E.g. the *mmpS4* and *mmpS5* single deletion mutants did not show any growth defect in low iron media [12], while the TnSeq results showed that MmpS4 is essential under all iron conditions examined in this study. However, in the TnSeq study we used the siderophore-deficient *mbtD* deletion mutant which prevented Mtb from utilizing iron salts as an iron source, whereas the isogenic mutants *mmpS4* and *mmpS5* single deletion mutants were grown in medium with iron salts [12]. When MBT was used as an iron source the *mmpL5* deletion mutant had a significant growth defect (Fig 7D) consistent with the importance of MmpL5 in the TnSeq results (Fig 5B).

## Does MmpL11 have a role in siderophore secretion?

Considering the proposed function of MmpL11 in heme uptake [24,58], it was surprising that the TnSeq data showed a significantly higher requirement for *mmpL11* in the presence of MBT by Mtb compared to hemin or hemoglobin (Figs 3B, 3D and 4; S3B Fig). Interestingly, *mmpL11* showed a transposon insertion pattern indicative of the siderophore-dependent self-poisoning phenotype (S3B Fig). The simplest explanation for this phenotype is that MmpL11 directly secretes siderophores similar to the related MmpL4 and MmpL5 efflux pumps. However, the Δ*mmpL4/L5* mutant did not grow at all in liquid medium indicating that MmpL4 and MmpL5 are sufficient for siderophore secretion in Mtb (Fig 7C and 7D). Alternatively, the known activity of MmpL11 in transport of surface lipids such as mycolic acid wax esters and long-chain triacylglycerols in Mtb and *M. smegmatis* [59,60] might indirectly influence the activity of the siderophore secretion systems of Mtb by lipid-protein interactions. Further investigation is clearly needed to clarify the role of MmpL11 in iron utilization by Mtb.

## Why do *mbt* genes not show a phenotype in TnSeq experiments?

Siderophore biosynthesis in Mtb is dispensable in the presence of either exogenous sidero-phores or heme [23,24]. The *mbt* genes encode enzymes required for siderophore biosynthesis [7,61] and were, as expected, not essential in our study using the Mtb Δ*mbtD* mutant with a defect in mycobactin synthesis [23]. Surprisingly, however, almost all *mbt* genes were classified as 'non-essential' in highly saturated transposon libraries on standard 7H10 medium containing copious amounts of iron salts [42]. The apparent permissiveness of *mbt* genes in TnSeq experiments is in conflict with the observation that Mtb cannot utilize iron salts and grow without siderophores [13] and with several reports demonstrating that the genetic or chemical inactivation of mycobactin biosynthesis results in severe growth defects of Mtb in medium with iron salts [23]. We reasoned that transposon insertions into *mbt* genes do not result in

growth defects of Mtb in TnSeq experiments because *mbt* mutants are cross-complemented by the many other clones with unaffected mycobactin biosynthesis in the pool of transposon mutants. Thus, the *mbt* genes should be included in the list of essential genes for utilization of iron salts and serve as a reminder of potential pitfalls of TnSeq experiments.

## Heme and hemoglobin utilization by Mtb

Most of the genes which are predicted by our TnSeq analysis to be required for optimal growth or are essential for heme utilization by Mtb are also required for optimal growth or are essential for hemoglobin utilization (Fig 4). This is consistent with our previous observation that both pathways appear to converge on the cell surface of Mtb [26]. However, the TnSeq analysis did not identify any gene required for this pathway such as the *ppe36* and *dpp* genes [26,36]. This might be a consequence of a second, previously unrecognized pathway for heme utilization which depends on albumin. Serum albumin binds heme with high affinity [62,63] and enables heme uptake independent of the *ppe36* and *dpp* genes, although these genes are essential for growth of Mtb in medium with heme or hemoglobin as sole iron sources in the absence of albumin [36]. Since Mtb was grown on agar plates containing albumin in our TnSeq experiments, albumin-dependent heme uptake might explain the lack of requirement for the *ppe36* and *dpp* genes. By contrast, *ppe37* which was identified in Mtb Erdman as required for heme utilization in the presence of albumin [27] was classified as "growth defect" in our TnSeq experiments. These divergent results might be caused also by significant strain differences between the Mtb H37Rv derivative ML1600 (ΔRD1 Δ*panCD* Δ*mbtD::hyg*[R]) used in our study (S1 Table) and the Mtb Erdman strain used in the previous study [27]. This hypothesis is supported by recent results demonstrating the deletion of *ppe37* in the H37Rv derivative Mtb mc$^2$6206 did not show any growth defect with heme as the sole iron source [36]. In this regard, it is noteworthy that transposon mutants with insertions in the genes encoding the heme oxygenase MhuD (Rv3592) which liberates iron by cleavage of the tetrapyrrole ring [64] and the heme-binding protein Rv0203 [65] did not show reduced fitness in the medium with hemin or hemoglobin (S8 Table). It is possible that the lack of phenotypes for some of these genes in the TnSeq experiments is caused by functional redundancy in Mtb or inter-strain complementation. However, strain-specific growth requirements or differences in growth conditions might also explain some of these divergent results.

The HMM analysis classified four genes encoding proteins of unknown functions as essential for utilization of hemin (*rv2016*, *rv3673c*) and hemoglobin (*rv0199*, *rv2420c*) (Fig 4, S8 Table). A detailed look into the insertion counts in the other biological replicates revealed that the *rv0199*, *rv2016*, and *rv2420c* genes still had a significant number of transposon insertions which suggesting that they are not truly essential under these conditions. However, the *rv3673c* gene had zero insertions indicating that it might indeed be involved in heme utilization by Mtb. The *rv3673c* gene was not essential when Mtb was grown on plates with siderophores. This classification is consistent with a previous TnSeq analysis when Mtb was grown in medium with iron salts [42]. Rv3673c is predicted to be a membrane-anchored thioredoxin-like protein. Further experiments with an isogenic mutant are needed to examine whether Rv3673c indeed plays a role in heme utilization by Mtb.

## Does heme poisoning exist in Mtb?

Considering the toxicity of excess heme for bacteria [66,67], we questioned whether our TnSeq analysis would also reveal genes with a heme poisoning pattern similar to that observed for siderophore poisoning, i.e. genes which are not essential in medium with cMBT or MBT, but essential or required for optimal growth in medium with hemin and with hemin+MBT.

Indeed, we identified seven genes with such a pattern: *rv0338c*, *rv0489*, *rv0542c*, *rv0548c*, *rv1122*, *rv3028c* and *rv3029c*. However, none of these genes encoded transport proteins. In particular we did not see such a phenotype for the efflux pump MmpL11 which was shown to be involved in heme utilization by Mtb [24]. Furthermore, Mtb grows very efficiently at hemin concentrations as high as 100 μM [19] indicating that Mtb is highly resistant to the toxic effects of heme in contrast to other bacterial pathogens such as *Staphylococcus aureus* which is strongly inhibited at 10 μM hemin [68]. Thus, it is questionable whether a heme poisoning effect exists in Mtb, i.e. protein-mediated uptake and intracellular accumulation of heme to toxic concentrations as it was shown for siderophores [13].

## Regulation of heme utilization by Mtb

The regulator protein Zur (Rv2359) was identified in our study as essential for utilization of heme by Mtb in the comprehensive TnSeq analysis and in growth experiments using an iso-genic Δ*zur* mutant. The dimeric Zur protein binds two Zn(II) ions per monomer [69]. Zn-binding by Zur is required to repress transcription of 32 genes in Mtb [50]. A straightforward explanation would be that Zur is an activator of genes required for heme utilization. However, no genes had lower transcription levels in a previous microarray analysis of an Mtb *zur* mutant compared with the wt strain indicating that Zur acts as a repressor [50]. It is conceivable that Zur represses a repressor gene which controls genes required for heme utilization. However, the only regulator gene under control of Zur appears to be *rv0232* [50] which encodes a TetR-like repressor which is not known to activate genes. The surprising essentiality of the *esx-3* operon for heme utilization by Mtb as shown in this study and the strong induction of the *esx-3* operon in the Mtb *zur* mutant [50] provide an alternative mechanism for the regulatory role of Zur in heme utilization. It is possible that the uncontrolled uptake of large quantities of heme mediated by the constitutively produced ESX-3 system in the absence of Zur is toxic for Mtb. The toxicity of excess heme is well documented in other bacteria [66,67,70,71] and might also require Mtb to synthesize PDIM and other lipids to "harden" the outer membrane as observed in our study (Fig 3B) and thereby prevent the uncontrolled diffusion of the hydro-phobic heme molecules into the cell. The apparent attempt of Mtb to control heme uptake on a transcriptional level is reminiscent of the important role of IdeR, the iron-dependent repres-sor of many genes involved in iron uptake and storage in Mtb [32,72] to prevent toxicity of free iron [1,4]. Obviously, further experiments are required to examine this hypothesis and to understand the role of the ESX-3 system and of the other genes of unknown functions revealed by this study in iron uptake from siderophores and heme.

## Conclusions

During much of the infection cycle Mtb resides inside host cells, mainly in macrophages but also in neutrophils [73]. Intracellular Mtb has access to iron from both heme- and non-heme proteins [74]. The available iron source is heavily shifted to heme in macrophages which recy-cle iron from damaged or senescent erythrocytes, mainly in the spleen [75]. During a chronic infection Mtb is localized in lung granulomas which contain extracellular bacteria [76,77]. A recent analysis of necrotic human granulomas revealed an abundance of host iron-sequester-ing proteins and iron-restricting factors, establishing an iron-deprived environment for Mtb [33]. This proteomic analysis also indicated that iron from both heme- and non-heme proteins is available to Mtb. Pulmonary tuberculosis is characterized by extensive tissue damage and airway bleeding, resulting in hemoglobin as the prevalent iron source. Thus, during most stages of infection Mtb has access to heme and non-heme iron sources. The much higher effi-ciency of siderophore-mediated uptake of iron ions indicates that this is the default pathway

primarily used by Mtb [13]. However, when heme becomes the major iron sources, Mtb switches from utilization of iron ions to heme. This switch appears to be mainly regulated by Zur and SmtB with an important role of the ESX-3 system and is accompanied by a metabolic shift involving more than 160 genes. Further studies are required to understand how this regulatory process is coordinated to enable Mtb to adapt to the iron sources available at different stages during infection.

# Materials and methods

## Bacterial strains, media, and growth conditions

All the strains used in this study are listed in S1 Table. The transposon mutant library was generated from Mtb Δ*mbtD*::*hyg*$^r$ (ML1600) [23] which derived from avirulent Mtb strain mc$^2$6230 (H37Rv ΔRD1 Δ*panCD*) [78]. For genetic manipulations, Mtb mc$^2$6230 derived strains were grown in Middlebrook 7H9 broth (Difco) supplemented with 10% ADS (50 g/L bovine albumin, 20 g/L dextrose, 8.5 g/L NaCl), 0.5% glycerol, 24 μg/mL pantothenate, 0.2% casamino acids and 0.01% tyloxapol. All transposon libraries were grown on self-made low-iron 7H9 Noble Agar (Difco) plates (supplemented with 10% ADS, 0.5% glycerol, 24 μg/mL pantothenate, 0.2% casamino acids, 0.01% tyloxapol, 20 μM 2,2'-dipyridyl (DIP) and 30 μg/mL kanamycin) with different iron sources: (i) 500 ng/mL Fe-cMBT; (ii) 250 ng/mL Fe-MBT (high concentration); (iii) 50 ng/mL Fe-MBT (low concentration); (iv) 20 μM hemin; (v) 5 μM hemoglobin and (vi) 20 μM hemin plus 250 ng/ml Fe-MBT, respectively. The siderophores cMBT and MBT were purified from *Mycobacterium bovis* BCG and *Mycobacterium smegmatis*, respectively, by chromatography and were provided by Dr. Colin Ratledge [79]. All Mtb plates were cultured at 37 °C for 21 days. Avirulent Mtb mc$^2$6206 and its derivative strains were grown in Middlebrook liquid 7H9 or solid 7H10 medium supplemented with 0.5% glycerol, 10% ADS, 0.2% casamino acids, 24 μg/mL pantothenate and 50 μg/mL L-leucine. *Escherichia coli* DH5α was grown in LB medium containing appropriate antibiotics at 37 °C with shaking at 200 rpm. The following antibiotics were used when required: kanamycin (Kan) at 30 μg/mL for mycobacteria or *E. coli*, and hygromycin (Hyg) at 200 μg/mL for *E. coli*, and 50 μg/mL for mycobacteria.

## Growth of Mtb strains in media with different iron sources

Mtb mc$^2$6230 (as wt) and Δ*mbtD*::*hyg*$^r$ (ML1600) were pre-grown in 7H9/hemin medium (Supplements: 0.5% glycerol, 24 μg/mL pantothenate, 0.2% casamino acids, 0.01% tyloxapol) until an OD$_{600}$ of 1.0 was reached. Then, the cells were iron-depleted in self-made low-iron 7H9 medium containing the supplements (less than 0.1 μM Fe$^{3+}$ as determined by ICP-MS) for 5–7 days. The iron-depleted cells were inoculated into 10 mL of low-iron 7H9 medium (containing the supplements) at an initial OD$_{600}$ of 0.01 with different iron sources: 20 μM ammonium ferric citrate; 10 μM human holo-transferrin; 10 μM human holo-lactoferrin; 5 μM human hemoglobin; 20 μM hemin; 0.2 μM Fe-MBT; 0.2 μM Fe-cMBT, respectively. The OD$_{600}$ of the cultures were determined every 24 hrs. All experiments were done in triplicate.

## Transposon mutagenesis

The Mtb Δ*mbtD*::*hyg*$^r$ (ML1600) strain was mutagenized by using *himar1*-based transposon as previously described with some modifications [39,40]. A total of 4.0×10$^{11}$ Mtb cells were harvested at OD$_{600}$ of 1.0 to 1.2. Cells were washed twice with MP buffer [50 mM Tris-Cl pH 7.5, 150 mM NaCl, 10 mM MgSO$_4$, 2 mM CaCl$_2$] and then infected with the ΦMycoMarT7 phage at an MOI of 20 at 37 °C for 4 hrs. Subsequently, the infected Mtb cells were washed twice with

10% glycerol and then cultured in self-made low-iron 7H9/ADS medium at 37 $^{o}$C for 24 hrs. The recovered cells were spread on self-made low-iron 7H9/ADS Noble agar plates containing the six iron sources as described above and 30 μg/mL kanamycin. Three independent libraries were generated for each of high Fe-MBT and hemin condition and two independent libraries were generated for each of the other four iron conditions. All plates were cultured at 37 $^{o}$C for 21 days. 14 independent libraries each containing ~200,000 transposon mutants were obtained.

## Preparation of chromosome-transposon junctions DNA for sequencing

Genomic DNA isolation, partial restriction digestion, ligation to asymmetric adapters, and transposon amplification were performed as previously described with modifications [39,40]. 12.5 μg library DNA was partially digested by *Hin*P1I (NEB). Fragments from 150 to 700 bp were purified from a 1.2% agarose gel using the Qiaquick gel purification kit (Qiagen). 2 μg *Hin*P1I-digested DNA was ligated with 8 μM adapter DNA (S2 Table) by T4 DNA ligase at 16 $^{o}$C overnight and the ligation products were purified using the Qiaquick PCR purification kit. The transposon-chromosomal junctions were selectively amplified twice by PCR. The first amplification was performed by the short transposon-specific PCR primers (S2 Table) with the PCR conditions: 96 $^{o}$C for 5 min; 20 cycles of 96 $^{o}$C for 1 min, 58 $^{o}$C for 1 min, 72 $^{o}$C for 45 sec, and 72 $^{o}$C for 2 min by LA-taq DNA polymerase (Takara). The PCR fragments from 200 to 600 bp were purified from a 2% agarose gel. The second amplification was performed using the hemi-nested PCR primers (S2 Table) with the following conditions: 96 $^{o}$C for 5 min; 20 cycles of 96 $^{o}$C for 1 min, 58 $^{o}$C for 1 min, 72 $^{o}$C for 30 sec, and 72 $^{o}$C for 2 min by LA-taq DNA polymerase. The PCR fragments from 250 to 450 bp were purified from a 2% agarose gel for Illumina sequencing. The libraries were subjected to 150-bp paired-end sequencing on the NextSeq 500 platform and raw sequence data were exported to fastq files for TPP processing.

## Processing of the TnSeq sequencing data

Raw reads were mapped to TA sites in the reference strain H37Rv (NC_000962.2) and read counts were reduced to unique template counts using a customized version of the TPP tool from version 2.0.2 of TRANSIT [45]: we modified the ADAPTER2 and three constant regions to match the sequences used in our protocol (Hendrickson RC, v2.0.2_zhang. 2019 Oct 4. In: GitHub [Internet]. Forked TRANSIT repository. Available from https://github.com/rusalkaguy/transit/tree/v2.0.2_zhang). The resulting .wig files containing the unique template counts were then processed with version 3.0.0 of TRANSIT [45] for statistical analyses. Template counts at each TA site were normalized by the TTR (total trimmed read count) method to adjust for differences in total counts between datasets [45].

## TnSeq analysis

The ANOVA (analysis of variance) method was used on the 14 independent datasets (S5 Table) grouped into the 6 iron conditions to determine which genes exhibit statistically significant variability ($\log_2$ fold-change, LTnSeq-FC) of the normalized Tn insertion counts across multiple conditions (using *f_oneway* in the *scipy* python package). *P*-values were adjusted for multiple comparisons using the Benjamini-Hochberg procedure. 165 genes ($\geq$ 3 TA sites) were identified (*q* values $<$ 0.05).

   The resampling analysis of TRANSIT (version 3.0.0) platform was performed on the triplicate samples (S5 Table) of high MBT and hemin conditions to compare the mutant frequency for each gene between the two conditions based on a permutation test as previously described [45]. The significant differences between the sum of the normalized read counts between the

high MBT and hemin conditions were evaluated by comparison to a null distribution derived by random permutation of the observed counts at TA sites. *P*-values were adjusted for multiple comparisons by the Benjamini-Hochberg procedure to control the false-discovery rate at $< 0.05$. Positive LTnSeq-FC or negative LTnSeq-FC values indicate an increased genetic requirement for utilizing MBT or hemin, respectively. The genes unrelated to either MBT or hemin are expected to have no significant difference (i.e. LTnSeq-FC values near 0).

The Hidden markov model (HMM) analysis of TRANSIT (version 3.0.0) was used on the biological replicate 1 dataset of each condition (S5 Table) to assess essentiality of the Mtb genes under different iron conditions as previously described [80]. The HMM models Tn insertion counts as coming from geometric distributions conditioned on four different states of essentiality with increasing mean insertion count: essential (ES), growth defect (GD), nonessential (NE), and growth advantage (GA). Genes are labeled based on the most frequent state label among individual TA sites in the gene (as assiged by the Viterbi algorithm). The numbers of genes in each state under different iron conditions are shown in S2 Fig.

### Construction of the *M. tuberculosis* deletion mutants

The Mtb deletion mutants Δ*rv2047c*, Δ*mmpL4*, Δ*mmpL5*, Δ*mmpL4/mmpL5* and Δ*smtB-zur* deletion mutants were constructed by allelic exchange using homologous recombination as previously described [81]. The parent strain Mtb mc²6230 harboring the *rv2047c* deletion vector pML3622 was grown on selective 7H10 plates containing 10% ADS, 0.5% glycerol, 24 μg/mL pantothenate, 0.2% casamino acids, 20 μM hemin, 2% sucrose and 50 μg/mL hygromycin. The plates were cultured at 40 °C for 4 weeks to select for double cross-overs (DCO). 6 out of 8 clones were validated as double cross-over (DCO) by PCR. One of the Δ*rv2047c::hyg^r* clones was selected and named Mtb ML2256 (S4 Fig). The *rv2047c* expression vector pML4211 was constructed and integrated into the chromosome of Mtb ML2256 at the L5 *attB* site to generate the *rv2047c* complemented strain Mtb ML2257. The deletion of the *smtB-zur* operon was done using the same protocol. The Mtb mc²6206 strain harboring the *smtB-zur* deletion vector pML3606 was grown on selective 7H10 plates at 40 °C for 4 weeks. 2 out of 3 clones were validated as double cross-over (DCO) by PCR. One of the Δ*smtB-zur::hyg^r* clones was selected and named Mtb ML2277 (S5 Fig). The L5 integration vectors pML4218 (*smtB*), pML4219 (*zur*) and pML4220 (*smtB-zur*) were transformed into Mtb ML2277 to generate the complemented strains Mtb ML2278, Mtb ML2279 and Mtb ML2280, respectively. All Mtb strains are listed in S1 Table and the deletion and expression vectors are listed in S4 Table.

### Supporting information

**S1 Fig. Heatmap of 165 *M. tuberculosis* genes showing statistically significant variability in transposon insertion counts across iron conditions.** ANOVA was used to identity the 165 genes which exhibited statistically significant changes in fitness under at least one of the tested iron conditions, after correction for false discovery rates. The mean normalized insertion count across replicates is calculated for each gene, for each of the 6 conditions. Then a log-fold-change is calculated for each condition relative to the mean count across all the conditions. The color 'red' means the counts in one condition are lower than the other conditions on average, suggesting a greater requirement for that gene in that condition, and 'blue' means insertion counts are higher than average, suggesting it is less required. The dendrogram shows the hierarchical clustering of the conditions (columns) using complete-linkage clustering. Gene insertion count profiles (rows) are also clustered using the *hclust* package in R, and gene pathway associations are indicated.
(TIF)

**S2 Fig. Quantitative classification of the requirement of genes by *M. tuberculosis* for growth under different iron conditions.** The essentiality of Mtb genes was classified using the HMM analysis. The number of genes in each class is indicated for each iron condition. (TIF)

**S3 Fig. *M. tuberculosis mmpL* genes involved in siderophore utilization in TnSeq.** Profiles of the transposon insertions in Mtb genomic regions including *mmpL4*, *rv0455c*, *mmpL5/S5* (A) and *mmpL11* (B). *MmpL3* is essential under all conditions consistent with previous results [82]. The iron conditions are indicated by different colors. The y-axis (0, 10, 20) represents the counts of the Tn-insertions and x-axis represents the genomic position (bp). Potential TA dinucleotide insertions sites are indicated in black. Regions containing genes of interest are boxed in red. Plots were generated using MochiView [46]. (TIF)

**S4 Fig. Construction of the *M. tuberculosis* Δ*rv2047c* deletion mutant.** (A) Schematic representation of the Mtb H37Rv *rv2047c* genomic regions and PCR performed to validate deletion of *rv2047c*. (B) PCR using primers (S3 Table) to validate *rv2047c* knock out (KO) mutants in the avirulent Mtb strain mc$^2$6230 (H37Rv ΔRD1 Δ*panCD*). (TIF)

**S5 Fig. Construction of the *M. tuberculosis* Δ*smtB-zur* deletion mutant.** (A) Schematic representation of the Mtb H37Rv Δ*smtB-zur* genomic regions and PCR performed to validate deletion of *smtB-zur*. (B) PCR using primers (S3 Table) to validate Δ*smtB-zur* knock out (KO) mutants in the avirulent Mtb strain mc$^2$6206 (H37Rv Δ*panCD* Δ*leuCD*). (TIF)

**S6 Fig. Growth of the *M. tuberculosis* Δ*smtB-zur* mutant and of the complemented mutant in iron-replete and iron-deficient media.** Growth assays of wild-type *M. tuberculosis* mc$^2$6206, ML2277 (Δ*smtB-zur* deletion mutant), ML2278 (Δ*smtB-zur* complemented with *smtB*), ML2279 (Δ*smtB-zur* complemented with *zur*) and ML2280 (Δ*smtB-zur* complemented with *smtB-zur*) under iron-replete (150 μM $Fe^{3+}$) 7H9 medium (A) and low-iron (less than 0.1 μM $Fe^{3+}$) 7H9 medium (B), respectively. The Mtb cells were grown in self-made low-iron 7H9 medium for 7 days to deplete intracellular iron before growth assays. The initial $OD_{600}$ of all the cultures is 0.01. Error bars represent standard deviations from the mean of results from biological triplicates. (TIF)

**S1 Table. Bacterial strains used in this work.** (PDF)

**S2 Table. Oligonucleotides used in this work for DNA sequencing.** (PDF)

**S3 Table. Oligonucleotides used in this work for construction of *M. tuberculosis* deletion mutants and for cloning of expression vectors.** (PDF)

**S4 Table. Plasmids used in this work.** (PDF)

**S5 Table. Statistical analysis of the *M. tuberculosis* transposon libraries used in this work.** (PDF)

**S6 Table. ANOVA analysis and functional categorization of genes involved in iron utilization by Mtb.**
(XLSX)

**S7 Table. Resampling analysis of genes involved in iron utilization by Mtb.**
(XLSX)

**S8 Table. Hidden Markov Model analysis of genes involved in iron utilization by Mtb.**
(XLSX)

**S9 Table. Shared genetic determinants of iron utilization by Mtb.**
(XLSX)

## Acknowledgments

We thank Dr. Eric Rubin (Harvard School of Public Health) for providing the ΦMycoMarT7 phage and Dr. Christopher Sassetti (UMass Medical School) for the TnSeq protocols and valuable advice. Purified siderophores were kindly provided by Dr. Colin Ratledge.

## Author Contributions

**Conceptualization:** Lei Zhang, Michael Niederweis.

**Data curation:** Lei Zhang, R. Curtis Hendrickson, Thomas R. Ioerger.

**Formal analysis:** Lei Zhang, R. Curtis Hendrickson, Thomas R. Ioerger, Michael Niederweis.

**Funding acquisition:** Elliot J. Lefkowitz, Michael Niederweis.

**Investigation:** Lei Zhang, Virginia Meikle.

**Methodology:** Lei Zhang, R. Curtis Hendrickson, Virginia Meikle, Thomas R. Ioerger.

**Project administration:** Lei Zhang, Virginia Meikle, Elliot J. Lefkowitz, Michael Niederweis.

**Resources:** Elliot J. Lefkowitz, Thomas R. Ioerger, Michael Niederweis.

**Software:** R. Curtis Hendrickson, Elliot J. Lefkowitz, Thomas R. Ioerger.

**Supervision:** Elliot J. Lefkowitz, Michael Niederweis.

**Validation:** Lei Zhang, Thomas R. Ioerger.

**Visualization:** Lei Zhang, R. Curtis Hendrickson, Thomas R. Ioerger, Michael Niederweis.

**Writing – original draft:** Lei Zhang, R. Curtis Hendrickson, Michael Niederweis.

**Writing – review & editing:** Lei Zhang, R. Curtis Hendrickson, Thomas R. Ioerger, Michael Niederweis.

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
