## [Decision Letter · Decision Letter 0]

2 Jan 2020

Dear Dr. Niederweis:

Thank you very much for submitting your manuscript "Comprehensive analysis of iron utilization by Mycobacterium tuberculosis" (PPATHOGENS-D-19-02065) for review by PLOS Pathogens. Your manuscript was fully evaluated at the editorial level and by independent peer reviewers. The reviewers appreciated the attention to an important topic but identified some aspects of the manuscript that should be improved.

We therefore ask you to modify the manuscript according to the review recommendations before we can consider your manuscript for acceptance. Your revisions should address the specific points made by each reviewer.

(1) A letter containing a detailed list of your responses to the review comments and a description of the changes you have made in the manuscript. Please note while forming your response, if your article is accepted, you may have the opportunity to make the peer review history publicly available. The record will include editor decision letters (with reviews) and your responses to reviewer comments. If eligible, we will contact you to opt in or out.

(2) Two versions of the manuscript: one with either highlights or tracked changes denoting where the text has been changed; the other a clean version (uploaded as the manuscript file).

We hope to receive your revised manuscript within 60 days or less. If you anticipate any delay in its return, we ask that you let us know the expected resubmission date by replying to this email.

[LINK]

Sincerely,

Helena Ingrid Boshoff

Associate Editor

PLOS Pathogens

JoAnne Flynn

Section Editor

PLOS Pathogens

Kasturi Haldar

Editor-in-Chief

PLOS Pathogens

orcid.org/0000-0001-5065-158X

Grant McFadden

Editor-in-Chief

PLOS Pathogens

orcid.org/0000-0002-2556-3526

The work is a valuable contribution to our understanding of iron acquisition pathways by the pathogen, M. tuberculosis giving a much broader and comprehensive understanding of genes that play a role in hemin or mycobactin mediated iron acquisition. There are some minor comments by all three reviewers - all of these will be easy to address in a revision and would serve to further improve the manuscript.

Reviewer's Responses to Questions

**Part I - Summary**

Reviewer #1: 1. The paper by Zhang et al discusses how iron is utilized in the human pathogen Mycobacterium tuberculosis (Mtb). This involves the accumulation of iron through interactions with secreted siderophores, mycobactin (MBT) and carboxymycobactin (cMBT), and can also use hemoglobin as a source of the heme iron. The nature of the paper is review-like in sections, although there are various important insights into the mechanisms of iron accumulation as well as descriptions of a transposon mutagenesis strategy linked with deep sequencing (TnSeq); work which led to the identification of 165 M. tuberculosis genes that are likely to be involved in iron uptake/utilization. The authors indicate that 66 of these genes are essential or needed for efficient Mtb cell growth in presence of sources of iron. The introduction provides a good coverage of what is currently known in relation to iron acquisition in Mtb, including a novel finding that the ESX-3 secretion system is required for heme utilization by Mtb, as well as for siderophore-mediated iron uptake. In the Results section the authors describe the construction of transposon libraries, with the Mtb transposon library grown on a complex mixture of compounds on agar plates, leading to libraries with large numbers of transposon insertions. New observations are made - including increased requirements for genes needed for lipid metabolism, PDIM production, the ESX-3 secretion system and leucine production in the presence of hemin or hemoglobin, but decreased requirements for other genes (e.g. for heme biosynthesis). The ppE4 gene was found to be non-essential for Mtb, Transposons in the IrtAB protein exhibited decreased "fitness" in the presence of MBT/cMBT. Other data included the finding that the MmpS4 protein likely has an unknown metabolic function in addition to its role in siderophore efflux. Further experiments revealed that the rv2047c gene is essential for Mtb growth to occur in the presence of high concentrations of MBT. Other results suggest that the MmpL4 and MmpL5 proteins are needed for siderophore secretion. The transcriptional regulators SmtB and Zur were shown to be essential for Mtb to grow in the presence of heme (20 microM). The Discussion revisits previous sections of the article and gives a good summary of the main findings in the article and their ramifications. Overall this is generally well written manuscript (albeit with a few typographical errors - see Part 3) and should probably be accepted subject to minor corrections.

2. The authors use a number of techniques in this manuscript. Of particular importance are the combined methods of high-density transposon mutagenesis and deep sequencing (TnSeq), which enabled the authors to make a series of observations that allowed them to make new insights into the functions of different enzymes in relation to iron utilization in Mtb.

3. There are some minor grammatical errors in the current version of the paper. However, the authors should be able to attend to these issues easily. The list is given below.

Reviewer #2: The authors have provided a detailed study on the differential utilization of iron sources by Mtb by exploiting the power of the TnSeq approach. This omics mutagenesis approach confirmed the essentiality of several genes and gene families that were previously shown to be important for Mtb uptake of iron via the mycobactin siderophores or by use of hemin or hemoglobin as a source of iron. Of importance the study identified a large number of genes that were only required for siderophore or heme based iron acquisition pathways. Thus, the results presented identified and tested several genes not previously shown to be essential for iron utilization via the siderophores or hemin/hemoglobin. The studies and results presented will serve as a major resource for the Mtb research community and provide a wealth of data for the development of new hypotheses and experimentation. The concerns that I have are minor.

Reviewer #3: The manuscript by Zhang et al. represents a thorough and valuable description of the genes necessary for M. tuberculosis to utilize different iron sources. Despite quite a bit of previous work, this remains an important topic. Not only are the mechanisms involved in the canonical mycobactin-dependent uptake pathway still incompletely defined, but those necessary for using heme-bound iron remain largely unknown. The authors use a TNseq approach to address this question, and produce a valuable and well-validated dataset. They identify many, if not all, of the known players in these pathways, as well as a number of novel genes that are somehow involved in these processes. While these novel genes were not investigated in mechanistic detail, the dataset provides some useful new insights into these processes, such as the reciprocal genetic requirements for mycobactin and heme utilization, which is a particularly striking finding. Both this dataset and the interpretation will prove valuable to the community.

**Part II – Major Issues: Key Experiments Required for Acceptance**

Reviewer #1: (No Response)

Reviewer #2: None

Reviewer #3: (No Response)

**Part III – Minor Issues: Editorial and Data Presentation Modifications**

Reviewer #1: 1) P.6, line 104 - presumably this should read "...were generated IN media containing...

2) P.7, line 129 - should read "...which exhibited A statistically significant....

3) P.8, line 157 - should read "...the electron transfer flavoprotein FixAB... (i.e. "flavoprotein" rather than "flavoproteins")

4) P.9, line 177 - should read "...previous experiments WHICH SHOW THAT neither the...."

5) P.9, line 198 - change "effect" to "affect"

6) P.15, line 306 - should read "...significant or complete growth defect of Mtb only.... (i.e. "defect" rather than "defects")

7) P.15, line 321 - should read "One example IS the hem genes..."

8) P.15, line 324 - should read "...standard Mtb growth MEDIUM containing iron salts..."

9) P.17, line 368 - delete the first "the" to read "...was confirmed by the absence of any growth defect..."

10) P.19, line 401 - should read "...experiment is in conflict WITH the observation that Mtb..."

11) P.19, line 413 - should read "...to converge on THE cell surface of Mtb [23]."

12). P.20, line 428 - correct the spelling of "tetrapyrrolE" (i.e. add an e on the end of the word).

13) P.21, line 451 - should read "...compared with THE wt strain, indicating that Zur..." (i.e. insert "THE" and insert a comma after "strain")

14) P.21, line 454 - should read "...repressor which IS not known to activate genes."

Reviewer #2: 1) Line 120-122, the authors imply that hemin rescues irtAB Tn mutants. However, the TN insertion profile of the library grown in hemin + MBT looks very similar to the profile of the TN library grown in MBT. Thus, it would seem that the irtAB could contribute to the MBT intoxification phenotype. How is the rescue of a phenotype quantified based on Tn insertion count?

2) Several genes are identified as being siderophore toxic secretion mutants, a concept that has been widely promoted in the Mtb research community. The identification of several genes that are previously reported to participate in efflux of MBT in these studies support this hypothesis. Were any genes identified that had a similar phenotype with hemin (i.e. low Tn insertion in hemin and hemin + MBT)

3) The authors identified a number of genes that lacked a defined function or a function not specifically associated with iron uptake or siderophore export. The authors do not consider that a proteins function may be directly tied to the source of iron. It would be interesting to know if gene products that require iron for function were identified in the differential iron source screen.

4) The results presented in Figure 7 and Page 12 did not include evaluation in the presence of hemin + MBT. This is a primary experiment to define a siderophore secretion phenotype mutant.

5) Line 104. I think the authors meant to say “generated on media” not “generated media.

6) Line 310 carboxymycobactin and mycobactin should be abbreviated

Reviewer #3: 1. Intro – A brief description of MBT vs cMBT might be helpful to non experts

2. Line 47-49 is this speculation or known?

3. Line 89 vs line 102 – be consistent with units even though plates and broth are different

4. Line 146-152 awkward wording

5. Line 201-239 – How is this section different from the next section? These should be merged or differentiated

6. Line 241 title is misleading in describing what is actually written in the section.

7. Line 299-301 – This text appears to associate smtB to hemoglobin and zur to heme but that is not supported by the data

8. Line 420. Have the authors determined how log heme is stable in agar? Perhaps heme degradation accounts for the discrepancies across different bodies of work?

9. Line 449-451. Under what condition was this done and was this relevant/comparable to iron-deficient conditions that this work examines?

10. The discussion could be improved by some attention to big biological questions on iron acquisition and how it impacts mtb pathogenesis. Like where would mtb preferentially use heme vs using siderophores?

11. Line 479 – how were these obtained?

12. Fig 6 and 7 – legend and figure are switched

13. Fig 7 –7D the mmpL5 growth lag is confusing. How this relates to the description that this gene is “required” should be discussed in more detail

PLOS authors have the option to publish the peer review history of their article (what does this mean?). If published, this will include your full peer review and any attached files.

Reviewer #1: No

Reviewer #2: No

Reviewer #3: No

---

## [Editor Report · Decision Letter 1]

20 Jan 2020

Dear Dr. Niederweis,

We are pleased to inform you that your manuscript 'Comprehensive analysis of iron utilization by Mycobacterium tuberculosis' has been provisionally accepted for publication in PLOS Pathogens.

Before your manuscript can be formally accepted you will need to complete some formatting changes, which you will receive in a follow up email. A member of our team will be in touch within two working days with a set of requests.

Best regards,

Helena Ingrid Boshoff

Associate Editor

PLOS Pathogens

JoAnne Flynn

Section Editor

PLOS Pathogens

Kasturi Haldar

Editor-in-Chief

PLOS Pathogens

orcid.org/0000-0001-5065-158X

Michael Malim

Editor-in-Chief

PLOS Pathogens

orcid.org/0000-0002-7699-2064

The authors have addressed all the reviewers' concerns (with the exception of the hemin + MBT experiment where the relevant strain was not available) and revised the manuscript accordingly. This work is an important contribution to our understanding of iron acquisition by M. tuberculosis presenting a genome-wide assessment of pathways required for utilization of this nutrient based on the sources available to it in the environment.
---

## [Editor Report · Acceptance letter]

5 Feb 2020

Dear Dr. Niederweis,

We are delighted to inform you that your manuscript, "Comprehensive analysis of iron utilization by *Mycobacterium tuberculosis*," has been formally accepted for publication in PLOS Pathogens.

Best regards,

Kasturi Haldar

Editor-in-Chief

PLOS Pathogens

orcid.org/0000-0001-5065-158X

Michael Malim

Editor-in-Chief

PLOS Pathogens

orcid.org/0000-0002-7699-2064